# Change detection of auditory tonal patterns defined by absolute versus relative pitch information. A combined behavioural and EEG study

**Nina Coy**[ORCID]*, **Maria Bader, Erich Schröger, Sabine Grimm**

Institute of Psychology–Wilhelm Wundt, Leipzig University, Leipzig, Germany

* nina.coy@uni-leipzig.de

**Data Availability Statement:** Data underlying the findings are deposited on the OSF repository: (10.17605/OSF.IO/PTGR3).

## Abstract

The human auditory system often relies on relative pitch information to extract and identify auditory objects; such as when the same melody is played in different keys. The current study investigated the mental chronometry underlying the active discrimination of unfamiliar melodic six-tone patterns by measuring behavioural performance and event-related potentials (ERPs). In a roving standard paradigm, such patterns were either repeated identically within a stimulus train, carrying absolute frequency information about the pattern, or shifted in pitch (transposed) between repetitions, so only relative pitch information was available to extract the pattern identity. Results showed that participants were able to use relative pitch to detect when a new melodic pattern occurred. Though in the absence of absolute pitch sensitivity significantly decreased and behavioural reaction time to pattern changes increased. Mismatch-Negativity (MMN), an ERP indicator of auditory deviance detection, was elicited at approximately 206 ms after stimulus onset at frontocentral electrodes, even when only relative pitch was available to inform pattern discrimination. A P3a was elicited in both conditions, comparable in amplitude and latency. Increased latencies but no differences in amplitudes of N2b, and P3b suggest that processing at higher levels is affected when, in the absence of absolute pitch cues, relative pitch has to be extracted to inform pattern discrimination. Interestingly, the response delay of approximately 70 ms on the behavioural level, already fully manifests at the level of N2b. This is in accordance with recent findings on implicit auditory learning processes and suggests that in the absence of absolute pitch cues a slowing of target selection rather than a slowing of the auditory pattern change detection process causes the deterioration in behavioural performance.

## Introduction

Acoustic information arriving at the human ear is fleeting and complex–in fact, auditory events are temporally and spectrally highly variable signals requiring efficient processing [1]. And yet, we can easily understand the same words articulated by speakers in different registers

**Funding:** The study was funded by the German Research Foundation (www.dfg.de) with the project number GR 3412/2-1. The funder had no role in study design, data collection and analysis, decision to publish, or preparation of the manuscript.

**Competing interests:** The authors have declared that no competing interests exist.

or volume, and recognise a melody played in another key or tempo. The meaningful sensory representation (i.e., a verbal message, a melody) derived from the, oftentimes ambiguous, acoustic input is typically referred to as an "auditory object". This term delineates "an acoustic experience that produces a two-dimensional image with frequency and time dimensions" [2]. Thus, despite being a general definition, it points out that the time course of spectral information provides salient cues about auditory events (e.g., a word, a short melody). Hence, it is not surprising that the human auditory system is finely attuned to the processing of spectral properties such as pitch at multiple levels of the central nervous system [3].

When a sound pattern occurs repeatedly, to extract and store its specific acoustic structure allows the auditory system to compare new input against this representation. In fact, the extraction of invariances facilitates the formation as well as the stabilisation of auditory representations [4], and their segregation from complex auditory environments [5].

In the simplest case, a specific auditory event recurs exactly within a natural auditory environment. For instance, when playing the same piano note twice, the spectral information is identical for both keystrokes. However, relevant auditory information (particularly in a sequence of auditory events) typically is not comprised in exact absolute spectral values but rather in their relative distances (e.g. pitch relations in music, formants in speech)–enabling the recognition of auditory objects despite large variation in spectral features [6–10]. As absolute spectral information provides only a limited basis for generalisation from prior listening experiences, it is of limited use outside highly familiar and rich contexts [11]. Actually, as absolute spectral information often is not available in learning situations, tolerance to absolute spectral variability must extend to the initial acquisition mechanisms. A first proof-of-principle was delivered by a study almost three decades ago, showing that not only first-order physical features of auditory stimuli are encoded in the brain, but also the relation between tone pairs can be derived from a series of varying physical events [12]. Since then, a rich body of research has emerged, enabling a better understanding of the processing of complex sounds [for review: 3, 13–16].

Indeed, a study published in 2017 found evidence that even in the absence of direct attention unfamiliar short melodic patterns are extracted when learning has to rely solely on relative pitch cues because absolute pitch information varies due to transpositions of these patterns [17]. In naïve listeners that performed a loudness change detection task, Bader and colleagues [17] observed a clear mismatch-negativity (MMN) in response to the introduction of a new pattern not only after previous exact (i.e. absolute) pattern repetitions but also when the previous pattern was transposed (i.e. relative) between repetitions. The MMN component of the event-related-potential (ERP) is typically observed as a frontocentral negative deflection in the deviant minus standard difference potential at 100–250 ms after deviation onset [18, 19]. MMN is considered to reflect the outcome of a process where a deviant event is found incongruent with "the predictions produced by the neural representations of regularities extracted from the acoustic environment" [16]. Bader et al. [17] observed that MMN was elicited after only three presentations of the preceding pattern and at a similar latency in both absolute and relative pitch contexts. Bader et al. [17] interpreted this as strong indication that the sensory memory trace forms automatically and violations of complex pattern regularities are extracted even without absolute pitch cues. This is in line with research showing that MMN is sensitive to higher order regularities [for review: 15, 19]. However, on the stimulus (not the difference) level deviant amplitudes increased significantly as a function of the number of preceding standard stimuli in the absolute pitch information context only. Bader et al. [17] suggested that this might be explained by a general attenuation of pattern change processing when reliant on relative pitch cues. Considering topographical differences between conditions, it also might indicate that different areas are involved in the processing of relative patterns.

Interestingly, P3a in response to the occurrence of new patterns was markedly affected in the form of a decreased amplitude and an increased latency by absence of absolute pitch cues. P3a is typically observed as a strong frontal/central positive deflection peaking at around 300–500 ms after deviation onset in response to task-irrelevant deviant stimuli [20–23]. Although it is not fully understood yet, which processes underlie the elicitation of P3a [24, 25], it has traditionally been associated with the (involuntary) orienting of attention towards the deviant auditory event [22, 26], more recently, with its evaluation [25, 27–29], and stimulus selection in working memory [24]. Bader et al. [17] interpreted the P3a modulation as a reflection of "the difficulty to distinguish implicitly between standard and deviant sound patterns in a relative pitch code context". This was supported during additional behavioural testing: when the emergence of new patterns was made explicitly task relevant, such pattern changes were detected less often and substantially slower when based on relative compared to absolute pitch cues.

One might argue that the observed P3a effects result from a substantial impoverishment of stimulus discriminability in the absence of absolute pitch. Nonetheless, it should not be neglected that the comparison of new patterns arguably is computationally more complex when reliant on relative compared to absolute pitch information, as it is not sufficient to compare whether two pitches are the same but rather whether their relative distance is. For instance, it was found that previously heard melodies that were in the same key at exposure and test were recognized with greater accuracy than melodies that were transposed [30–33], and that the ability to process relative pitch information depends on experience [30, 32, 34–37]. In our case, when absolute pitch information is available, any change in pitch is indication enough that the listener hears a new pattern. Deviant relative patterns can only be identified as such by comparing relative distances between at least two pitches within a pattern in relation to the previous stimuli. Thus, the P3a latency effect might be attributable to differences in computational processing demands. Actually, this might also concord with different brain areas being involved, which fits well with the observations described above about the topography of the deviant response at the level of MMN. Nevertheless, the consistent latencies of MMN in the two contexts rather confirm that the auditory discrimination process involved in detecting a pattern change is not necessarily delayed in the absence of absolute pitch cues. Thus, the considerable P3a differences might be (partly) attributable to an uncertainty at higher levels about which auditory changes are actually relevant in a passive listening situation [24]. Though, P3a might reflect stimulus evaluation and not mere orienting of attention toward the deviant stimulus, that does not mean it is independent from attentional modulation [22, 27, 38, 39]. Whereas a new pattern in a context of identical pattern repetitions constitutes a clear deviation to the preceding stimulation and elicits P3a [17, 40], the occurrence of a new pattern in a relative pitch code context must not necessarily be inherently more relevant to the brain than if it were a transposition of the preceding pattern–in both cases absolute pitch information has changed. For instance, when a word is spoken first in a low and then in a higher register it might indicate emotional arousal in a person (i.e., a change within the current source) just as well as it might signify that another speaker simply repeated what the first said (i.e., change to a new source). Also, it might not be adaptive to shift resources to every change (i.e., transpositions, pattern and loudness changes) within a high change environment (i.e., relative pitch context) which holds no task-relevant information for a listener. Therefore, the amplitude difference is possibly accounted for by bottom-up attentional modulation.

In sum, these findings indicate a striking divergence between the automatic change detection regardless of absolute pitch information at the level of MMN on the one hand, and the decreased further evaluation processing at the level of P3a. This poses the question whether the P3a amplitude decrease and latency increase reported by Bader et al. [17] reflect a difference in stimulus discriminability, computational complexity, bottom-up relevance, or a combination

of these factors as a function of availability of relative compared to absolute pitch information. Although Bader et al. [17] observed that the discrimination performance between pattern repetition and true pattern change was significantly decreased by lack of absolute pitch information, one should consider the following: in the instances that a new pattern based on relative pitch information was correctly identified as such, still a prolongation of response latency occurred, it seems unlikely that the P3a effect is only a question of relevance. The aim of the experiment on hand was to investigate this temporal delay as well as a possible amplitude modulation on the electrophysiological level in a setting, in which the occurrence of new patterns is top-down relevant. That means, participants are explicitly asked to detect pattern changes to ensure that they actively attend to deviant patterns.

Other studies employing active discrimination paradigms have shown that the components MMN, N2b and the P300 complex typically characterise the ERP in such situations [38, 41]. It has also been reported that in active paradigms MMN can be difficult to estimate due to a temporal and spatial superposition of N2b [18, 19, 42]. The generic N2b component usually shows at approximately 200–350 ms after stimulus onset with a modality specific topography–in response to auditory stimuli it manifests as a central negative deflection [43, 44]. It has been related to intentional higher-order processing of change (mismatch) and of task-relevant (match) stimulus characteristics [43, 45–47]. Dien, Spencer & Donchin (2004) have discussed N2 as a process operating on a stimulus identification stage, though Ritter et al. (1983) have argued that it despite reflecting a comparison process between current stimulus and a voluntarily held target template, presence of N2b alone should not simply be equated with the actual detection of a target. The aforementioned P3a actually often occurs within a broad positivity at around 300–500 ms after stimulus onset observed in stimulus discrimination contexts, termed P300-complex [22, 23, 48]. In response to target stimuli the P300 complex also includes the more posterior P3b [21–24, 48]. Estimation of components within the P300 range often is difficult when there is a temporal overlap of central P3a and parietal P3b, sometimes adding up to one big late positive potential [21, 24, 49, 50]. P3b is typically assumed to index the revision, or updating of the current mental model of the stimulus environment in response to an unexpected task-relevant event [51, 52]. The postulation that P3b is largely independent from response selection and execution processes [51, 53, 54] has been challenged though [49, 55, 56], as has the notion that P300 elicitation warrants unexpectedness [22, 24].

In general terms, P300 is associated with (focal) attention and (working) memory operations [21–24, 57]. Though, it is not entirely clear whether P3a represent the same underlying processes in passive and active listening situations [24].

To summarise, MMN is mainly associated with automatic (non-intentional), P3a with somewhat semi-automatic processing, while N2b and P3b are related to intentional processing. Therefore, these four components and their relation to the behavioural output might offer insight into how, or when auditory processing is affected when absolute information about abstract patterns is variable. Extrapolating from the findings by Bader et al. [17], we hypothesise an impairment of behavioural performance visible in decreased pattern discrimination accuracy as well as prolonged reaction times in the absence of absolute pitch cues. Furthermore, that automatic auditory pattern processing (MMN) remains relatively unaffected by variability of absolute pitch information, but processes associated with direct attention (N2b, P3b) are more vulnerable to the absence of absolute pitch information. It is not really clear whether this is also true for P3a, or whether the explicit relevance of true pattern changes can compensate [24] the P3a impoverishments observed in the passive listening situation by Bader et al. [17] in a relative pitch context.

## Methods

### Participants

Data was collected at Leipzig University, protocol and procedures were in accordance with the Declaration of Helsinki and approved by the ethics committee of the Medical Faculty at Leipzig University (Az: 089-15-09032015). Participants either received credit points or were paid in compensation for their collaboration. Seventeen healthy people (18–31 years, 10 female) participated in the experiment on hand. Two additional participants were tested but excluded due to extensive artifacts. All participants reported normal hearing, and normal or corrected-to-normal vision. Self-reported musical expertise varied between no experience and 14 years of having played an instrument and, or sung in a choir ($M = 5.79$; $SD = 4.88$) but none of the participants were professional musicians.

**Procedure and apparatus.**   Participants were seated comfortably in a soundproof chamber during the experiment. The auditory stimuli were created in MATLAB R2013a (MathWorks, 2013) and presented with Psychtoolbox [58] binaurally via headphones (HD25-I 70 Ω, Sennheiser GmbH & Co.KG, Germany) at approximately 73 dB sound pressure level. Participants were instructed to listen to melodic sound patterns. Before each block they were instructed via a computer screen, approximately 100 cm away from their eyes, to respond as quickly and as accurately as possible with a button press, whenever they noticed a pattern being genuinely different from the preceding one, i.e., not a mere transposition. Behavioural responses (button press) were registered via reaction time box (Response Time Box, Suzhou Litong Electronic Co., Ltd., China). Feedback of performance was given on the screen after each block. During the auditory stimulation, participants were asked to gaze at a white fixation cross against a black background at the centre of the screen. The whole experiment, including preparation and two breaks, took approximately three and a half hours.

### Stimuli and design

Each auditory pattern was 300 ms in duration and composed of six seamlessly concatenated segments of 50 ms duration. Each segment was a compound of a fundamental tone, drawn randomly from within the frequency range between 220–880 Hz (i.e., the range of an octave), and several harmonics (decreasing in intensity with a linear slope and a cut-off at 6000 Hz). All segments included 5 ms rise and fall time, intensity was root-mean-square normalised.

The pattern change detection task was conducted in the form of a roving standard paradigm [17, 59, 60]. A randomly generated pattern was presented either 2, 3, 5 or 8 times successively (stimulus train). The last respective presentation of a given pattern served as the *standard* stimulus, and the first pattern of the subsequent stimulus train, which was incongruent with the pattern from the previous stimulus train, as a *deviant* stimulus. Therefore, there was no constant standard or deviant stimulus throughout a block.

In the absolute condition (ABS) a pattern was repeated identically within its stimulus train, thus each repetition consisted of the exact same configuration of pitches. Whereas, in the transposed condition (TRA) only the relations between pitches of a sound pattern were repeated by shifting the whole pattern by at least one semitone up or down in pitch (i.e., transposition over 12 equidistant semitone steps)–absolute pitches were thus changed between pattern presentations within a stimulus train. For an exemplary illustration of a pattern sequence please refer to Fig 1 and for a listening example to S1 and S2 Files.

There were 800 ms of silence between each pattern presentation. As at least two segments were necessary to discriminate a transposed pattern from the previous one (relation between the first two tones), the first segment of absolute patterns was fixed at 400 Hz so that

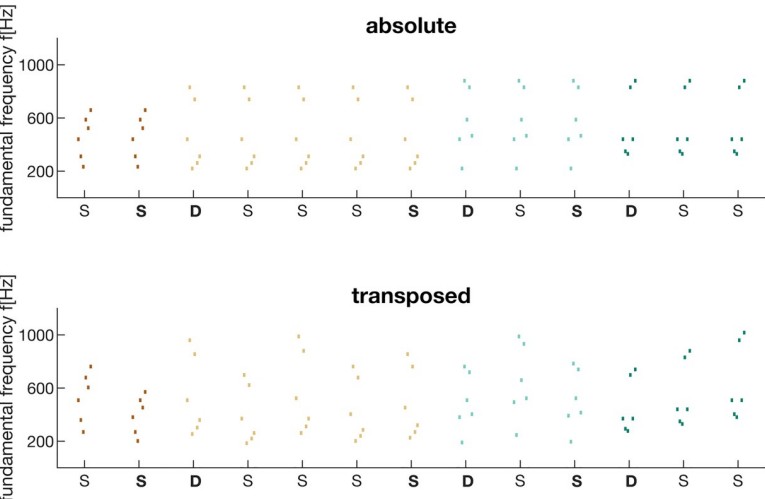

**Fig 1. Exemplary sequence of auditory patterns presented in a roving standard paradigm.** Each pattern is composed of six seamlessly concatenated 50 ms segments. Each segment was comprised of a fundamental, randomly drawn from 220–880 Hz, and added harmonics with a slope until 6000 Hz. Note, that for simplicity only fundamentals are depicted in the stimulus illustration. Sound patterns were presented for a certain number of times [2, 3, 5, 8] with a constant SOA (1100 ms), until a new pattern was introduced. The last pattern in each train served as standard (S), while the first presentation in each new pattern train was defined as a deviant (D) pattern (i.e., true pattern change). In the absolute condition (*top*, cf. S1 File) patterns within a stimulus train were repeated identically, carrying absolute frequency information about the pattern. In the transposed condition (*bottom*, cf. S2 File) patterns as a whole were shifted up or down in pitch (transposed) with a train, thus only relative frequency information was available to extract pattern identity. Participants were instructed to press a button whenever they detected a genuine pattern change (i.e., ignore transpositions).

congruently in this condition only the second segment indicated a potential pattern change. Each train length occurred ten times in a randomised order within each block. There were 10 blocks (400 deviants) of the absolute and 14 blocks (560 deviants) of the transposed condition, in order to compensate for the expected reduced hit rate of relative pattern deviants.

## Data acquisition

Reaction time (RT) was defined as the time between pattern onset and button press, as long as it did not exceed the inter-stimulus-interval. In accordance with signal detection theory, a button press in response to a deviant pattern (target) was treated as a hit, whereas a button press in reaction to a standard pattern (non-target) was registered as a false alarm. Discrimination sensitivity (d') between standard and deviant patterns was estimated using the log-linear correction [61].

EEG data was recorded continuously at 64 Ag/AgCl active electrodes, mounted according to the 10–10 international system in a suitable head cap, and amplified with a BrainAmp DC (Brain Products GmbH, Germany) amplifier and digitised with a sampling rate of 500 Hz. Eye movements (EOG) were measured horizontally with two electrodes positioned at the outer canthi of the eyes and vertically with one electrode below the left eye which was bipolarised with Fp1. Impedances were kept below 5 kΩ at all electrodes.

**Data processing.** EEG data analysis was performed offline in MATLAB (MathWorks, R2018b) and with the EEGLAB toolbox [62]. Data were consecutively filtered [63, 64] using a 0.1 Hz high-pass filter (Kaiser windowed sinc FIR filter, order = 9056, beta = 5.653, transition bandwidth = 0.2 Hz) and a subsequent 35 Hz low-pass filter (Kaiser windowed sinc FIR filter, order = 364, beta = 5.653, transition bandwidth = 5 Hz). Noisy channels, except EOG channels,

with a $z$ standardised standard deviation greater than 4 were removed from further analysis. The data were segmented into epochs of 900 ms in duration time-locked to the stimulus onset and included 100 ms baseline. Trials were excluded if maximal peak to peak difference was greater than 750 µV. Independent component analysis (ICA), using an infomax algorithm implemented in the pop_runica function of EEGLAB, was applied to further correct the data for artifacts with [65]. To improve decomposition, ICA was computed (exclusive of bad channels and trials) on the 1 Hz high-pass (Kaiser windowed sinc FIR filter, order = 3624, beta = 5.653, transition bandwidth = 0.5 Hz) and subsequently 35 Hz low-pass (see above) filtered raw data. To shorten computation time, the data were down-sampled to a 250 Hz sampling rate. ICA weights were then transferred onto the 0.1–35 Hz data. Artifactual components were semi-automatically identified [66] using the EEGLAB plugins SASICA, ADJUST [67] and FASTER [68], and subsequently removed from the data. Previously excluded channels were spherically spline interpolated [69] afterwards. Epochs were baseline corrected using the 100 ms before stimulus onset. Trials exceeding a 100 µV peak-to-peak difference were excluded from analysis. For each subject the remaining trials (epochs per cell: $M$ = 291; $SD$ = 108) were averaged for each stimulus type (standard and deviant) in each condition (absolute and transposed). Grand averages were computed from these subject-level ERPs, and difference waves by subtracting ERPs in response to standards from those to deviants.

Amplitudes were extracted on the subject level by window-averaging around each component's respective grand-average peak (window widths: MMN: 40 ms, N2b: 60 ms, P3a: 40 ms, P3b: 100 ms) for each stimulus in both conditions within either an anterior (MMN, N2b and P3a) or a posterior (P3b) 3x3 electrode cluster. The jackknife-scoring method [70] was used to estimate the time course of each ERP component at a respective electrode of largest activation (MMN: Fz, N2b: FCz, P3a: FCz, P3b: Pz). Specifically, the time point was estimated, at which the amplitude of a particular component across leave-one-participant-out subsamples of the grand-averaged wave first reaches specific percentage values of the respective peak amplitude; slope (60%) and peak (100%). The 60%- relative peak estimate was included firstly, because relative latency estimates have been shown to be less noisy than peak latency estimates using the jack-knifing technique, and secondly [70], to probe whether latency effects are already present in the build-up of a given component. The search windows for jackknife estimation were defined based on the grand-average to avoid overlap of components as follows: 100–230 ms for MMN, 180–500 ms for N2b, 250–550 ms for P3a, and 300–800 ms for P3b.

## Statistical analysis

The statistical analysis was conducted in RStudio Desktop Open Source Edition Version 3.6.2 [71], with *ez* package [72], *multtest* package [73], and *ggplot2* package [74].

Reaction times and sensitivity indices were tested between conditions (absolute vs. relative) by means of a paired student's *t* test.

The reported *t* values of the jackknife latencies comparing between absolute and transposed condition, as well as corresponding standard errors were adjusted to correct for an artificial reduction in error variance induced by jackknifing [75, 76]. Differences in amplitudes were statistically analysed by means of a 2x2x3 repeated measures ANOVA for each component respectively with the within-subject factors condition (absolute vs. transposed) x stimulus type (standard vs. deviant) x frontality [MMN, N2b, P3a: anteriority (frontal, frontocentral, central) / P3b: posteriority (parietocentral, parietal, parietooccipital)]. As there were no meaningful effects of laterality, the lateral dimension was collapsed, in order to simplify the statistical analysis. Please note that the activity values along the midline (factor frontality) represent averaged values not only including the central electrode but also the respective lateral electrodes directly

adjacent to the midline electrode; e.g., the factor level frontal is the average of Fz (middle), F3 (left) and F4 (right).

In case of violations of the sphericity assumption, Greenhouse-Geisser corrections were applied, and corrected *p* values are labelled with "GG". Post hoc analysis of significant interactions was conducted by means of within factor level repeated measures ANOVAs and post-hoc paired student *t* test. To investigate presence of the components of interest, deviants were compared against standards respectively, to probe for condition effects on these components the magnitude of the deviant minus standard differences were compared between conditions by means of post hoc paired student's *t* tests. In multiple pairwise comparisons the two-step Benjamini-Hochberg procedure [77] was applied to control the false discovery rate at a level of 5%, which has been shown to yield a good trade-off with statistical power [78]; in these instances only the adjusted *p* values are reported.

Data were deposited in the OSF repository. https://doi.org/10.17605/OSF.IO/PTGR3 [79].

## Results

### Behavioural results

As can be seen in Fig 2, participants detected true pattern changes well above chance even in the absence of absolute pitch. Nonetheless, discrimination performance is significantly decreased (difference in d': $M$ = -2.34; $SD$ = 0.57, see Table 1) when only relative pitch cues are available, $t(16)$ = -16.983, $p < .001$, $d$ = -4.119. The available pitch information explains approximately 81% of the variance in the discrimination performance ($\eta^2$ = 0.808). On average participants take 67 ms ($SD$ = 23 ms) longer to press the button in response to a true pattern

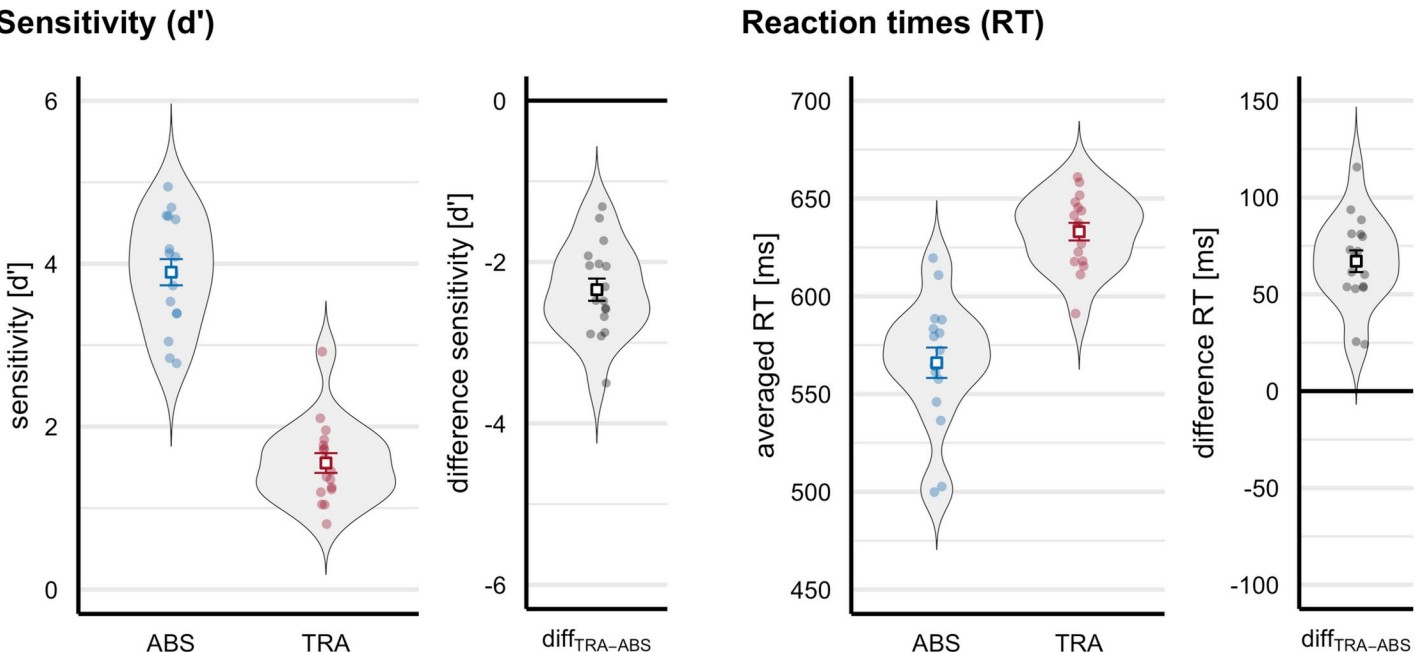

**Fig 2. Violin plots depict behavioural performance measures as a function of condition.** The plots show (n = 17 participants) sensitivity (d') in discriminating true pattern changes from pattern repetitions (*left panel*), and averaged reaction times (RT) when correctly responding to true pattern changes (*right panel*). The absolute condition is depicted in blue, the transposed in red and the respective absolute–transposed difference in black. Small dots represent each participant's mean performance, the bold square the group average and error bars ± 1 SEM. Participants detected true pattern changes well above chance even in the absence of absolute pitch. Nonetheless, sensitivity to true pattern changes based on relative compared to absolute pitch information is lower (d': $M$ = -2.34) and the behavioural response on average 67 ms slower respectively.

**Table 1. Sensitivity index (d'), reaction times (RT in ms), jackknife latencies (ms) and window-averaged amplitudes (µV) of the deviant minus standard difference components MMN, N2b, P3a and P3b in the absolute and the transposed condition.**

| Measure | EP | absolute | | | | transposed | | | | tra vs. abs | |
|---|---|---|---|---|---|---|---|---|---|---|---|
| | | std | dev | $\Delta_{dev,std}$ | *d* | std | dev | $\Delta_{dev,std}$ | d | diff | *d* |
| **d'** | | | | 3.90 | | | | 1.55 | | -2.34 | **-4.119** |
| **RT** | | | | 566 | | | | 633 | | 67 | **2.906** |
| **MMN** | | | | | | | | | | | |
| latency | Fz | | | | | | | | | | |
| slope | | | | 209 | | | | 172 | | -36 | -0.088 |
| peak | | | | 230 | | | | 206 | | -24 | -0.098 |
| amplitude | F | -1.54 | -2.77 | -1.22 | **-0.97** | -0.40 | -0.92 | -0.52 | **-0.55** | 0.94 | **0.493** |
| | FC | -1.17 | -2.53 | -1.36 | **-1.16** | -0.39 | -0.72 | -0.34 | **0.34** | 0.70 | **0.698** |
| | C | -0.50 | -1.72 | -1.22 | **-1.23** | -0.41 | -0.49 | -0.08 | **-0.08** | 1.02 | **0.820** |
| **N2b** | | | | | | | | | | | |
| latency | FCz | | | | | | | | | | |
| slope | | | | 219 | | | | 293 | | 74 | **0.797** |
| peak | | | | 288 | | | | 339 | | 51 | **1.703** |
| amplitude | F | -2.77 | -4.66 | -1.90 | **-0.98** | -2.84 | -3.99 | -1.15 | **-0.82** | 0.74 | -0.413 |
| | FC | -2.36 | -4.18 | -1.82 | **-0.84** | -2.68 | -3.70 | -1.01 | **-0.66** | 0.80 | -0.402 |
| | C | -1.60 | -2.74 | -1.14 | **-0.61** | -2.48 | -2.83 | -0.35 | **-0.20** | 0.79 | -0.433 |
| **P3a** | | | | | | | | | | | |
| latency | FCz | | | | | | | | | | |
| slope | | | | 422 | | | | 452 | | 30 | 0.207 |
| peak | | | | 489 | | | | 492 | | 3 | 0.030 |
| amplitude | F | -2.77 | -1.11 | 1.66 | **0.43** | -2.95 | -1.57 | 1.39 | **0.43** | -0.28 | -0.091 |
| | FC | -2.03 | 0.82 | 2.85 | **0.75** | -2.38 | -0.30 | 2.07 | **0.65** | -0.77 | -0.232 |
| | C | -1.26 | 3.90 | 5.16 | **1.35** | -1.83 | 1.71 | 3.55 | **1.17** | -1.62 | -0.447 |
| **P3b** | | | | | | | | | | | |
| latency | Pz | | | | | | | | | | |
| slope | | | | 420 | | | | 514 | | 94 | **0.857** |
| peak | | | | 630 | | | | 670 | | 40 | 0.323 |
| amplitude | PC | -0.31 | 9.10 | 9.40 | **2.17** | -0.84 | 8.17 | 9.01 | **2.17** | -0.39 | -0.098 |
| | P | 0.03 | 10.73 | 10.70 | **2.34** | -0.44 | 10.22 | 10.66 | **2.40** | -0.04 | -0.010 |
| | PO | 0.49 | 10.52 | 10.03 | **2.54** | 0.40 | 10.91 | 10.51 | **2.42** | 0.48 | 0.145 |

*Note*. EP: electrode positions, F: frontal (F3, Fz, F4), FC: frontocentral (FC3, FCz, FC4), PC: parietocentral (PC3, PCz, PC4), P: parietal (P3, Pz, P4), PO: parietooccipital (PO3, POz, PO4). *d*: Cohen's $d_z$. Significant differences ($p$ or $p_{adj} < .05$) are printed in bold.

change in the transposed compared to the absolute condition, $t(16) = 11.980$, $p < .001$, $d = 2.905$. The available pitch information explains approximately 64% of the variance in the reaction times ($\eta^2 = 0.636$).

## EEG results

For a rough characterisation, grand-average peak latency values are reported, but please note that jackknife latencies will be used for statistical analysis. The grand-averaged deviant-minus-standard difference waveforms are characterised by an initial negative going deflection (Fig 3A). In the transposed condition there is a first negative difference at Fz which is maximal around 185 ms (MMN) and is followed by a second negative difference peaking at 342 ms (N2b). However, in the absolute condition the negative difference rises to one prominent peak

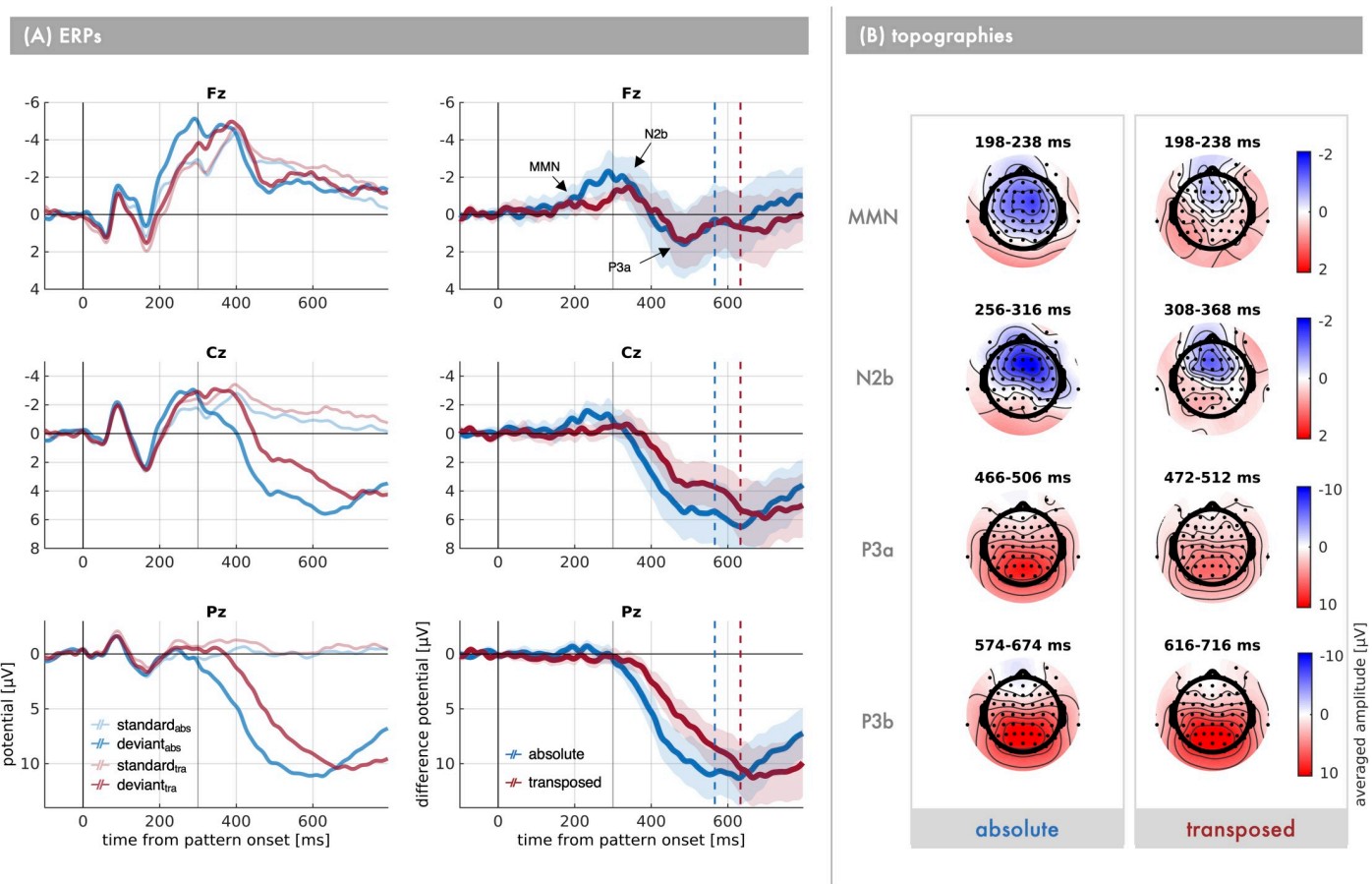

**Fig 3. Grand averaged ERP waves.** (A) Grand-averaged nose-referenced ERP waves of n = 17 participants at midline electrodes Fz (top), FCz (middle) and Pz (bottom) as a function of *condition* (blue: absolute, red: transposed) and *stimulus* (left: standard and deviant response, right: deviant–standard difference). The shaded area represents ± 1 SEM around the difference potential. The ERP components are labelled where appropriate. The solid grey vertical line marks the pattern offset, while the dashed vertical lines denote the average response time in each condition. Please note, that the point of deviation occurs at 50 ms, that is with the start of the second segment of a pattern. (B) Topographies depict the window-averaged deviant–standard difference amplitude around each ERP component's peak for both conditions (left: absolute, right: transposed) respectively.

at 286 ms, indicating a potential overlap of MMN and N2b activity. In both conditions the negativity of the deviants subsequently decreases, resulting in a positive difference (P3a) visible most clearly at anterior electrodes–maximal at FCz at 488 ms in the absolute and 490 ms in the transposed condition. Simultaneously, a strong posterior positivity (P3b) of the deviants starts to build up and is maximally different from the standards at 626 ms and 666 ms, respectively, at Pz. Principally, the ERP waves in response to both standard and deviant pattern stimuli appear morphologically quite similar between the absolute and transposed condition, though visually not identical with regard to the time course and magnitude of the deviant minus standard difference.

To test for effects of pitch information (condition) on topography (frontality) and magnitude of window averaged amplitudes associated with pattern repetitions and true pattern changes (stimulus), a repeated measures ANOVA was computed including the factors and their interactions (full model): *condition* (absolute vs. transposed) x *stimulus type* (standard vs. deviant) x *frontality* [MMN, N2b, P3a: anteriority (frontal, frontocentral, central) / P3b: posteriority (parietocentral, parietal, parietooccipital)].

**MMN.** Topographies of the averaged difference amplitude values within the MMN time window (Fig 3B) show that the initial negative difference between deviants and standards is mainly distributed at anterior sites with an inversion at more posterior, including the mastoid electrodes. In the transposed condition it is concentrated mostly at frontal electrodes, whereas in the absolute condition it shows a broader distribution centred around frontocentral electrodes with a slight right-hemispheric tendency.

*Latencies.* Jackknife latencies were extracted within a search window between 100 ms and 230 ms. In the transposed condition 60 percent of the peak amplitude of that first negative component are reached at approximately 172 ms ($SE$ = 77 ms) at electrode Fz (Fig 4A), peaking at 206 ms ($SE$ = 66 ms) which is after the fourth segment of the pattern ended. The estimated jackknife latencies did not differ significantly from the absolute condition, slope: $t_{adj}(16)$ = 0. 404, $p_{adj}$ = .515, $d$ = 0.098; peak: $t_{adj}(16)$ = 0.364, $p_{adj}$ = .515, $d$ = 0.089. However, in the absolute condition, visually, there is no clear peak in this time frame. In fact, the jackknife peak latency value in the absolute condition ($M$ = 230 ms; $SE$ = 0 ms) corresponds to the upper limit of the search window for the MMN peak. Thus, the slope latency in the absolute condition ($M$ = 209 ms; $SE$ = 38 ms) actually reflects the time point at which sixty percent of the amplitude at the upper search boundary are reached. In order to verify that this issue did not confound our results, we further compared the latencies at which 60% of the peak amplitude in the transposed condition were reached in both conditions respectively (absolute: $M$ = 209 ms; $SE$ = 5 ms; transposed: $M$ = 238 ms; $SE$ = 56 ms), again yielding no significant difference in the MMN slope, $t_{adj}(16)$ = -0.513, $p$ = .615, $d$ = -0.124. In sum, there is no indication of a delay in MMN build-up as a function of pitch information.

*Amplitudes.* Within the MMN time range, activity independent of stimulus generally increases in negativity from central towards frontal electrodes (Fig 4B), main effect of anteriority: $F(2,32)$ = 7.031, $p_{GG}$ = .013, $\eta_g^2$ = 0.009. Activity in response to the auditory stimuli is generally more negative in the absolute compared to the transposed condition, main effect of condition: $F(1,16)$ = 27.035, $p$ < .001, $\eta_g^2$ = 0.042. Also, the distribution of activity within the MMN window differs between conditions, condition*anteriority interaction: $F(2,32)$ = 24.870, $p_{GG}$ < .001, $\eta_g^2$ = 0.004. Only in the absolute condition negative amplitudes increase from central to frontal electrode sites (anteriority$_{abs}$: $F(2,32)$ = 17.639, $p_{GG}$ < .001, $\eta_g^2$ = 0.026), while this effect is absent in the transposed condition (anteriority$_{tra}$: $F(2,32)$ = 0.703, $p_{GG}$ = .432, $\eta_g^2$ = 0.001;).

More interestingly, amplitudes in response to deviant patterns are significantly more negative compared to amplitudes for standard patterns, main effect of stimulus: $F(1,16)$ = 18.475, $p$ < .001, $\eta_g^2$ = 0.020. Firstly, this main effect is dependent on the condition, condition*stimulus interaction: $F(1,16)$ = 8.019, $p$ = .012, $\eta_g^2$ = 0.008. Secondly, the stimulus main effect is further characterised by a condition*stimulus*anteriority interaction: $F(2,32)$ = 5.879, $p_{GG}$ = .015, $\eta_g^2$ = 0.0003. Post hoc analysis revealed that in the absolute condition there are significant additive but no interaction effects of stimulus and anteriority (stimulus$_{abs}$: $F(1,16)$ = 22.814, $p$ < .001, $\eta_g^2$ = 0.050; anteriority$_{abs}$: $F(2,32)$ = 17.639, $p_{GG}$ < .001, $\eta_g^2$ = 0.025; stimulus*anteriority$_{abs}$: $F(2,32)$ = 0.600, $p_{GG}$ = .484, $\eta_g^2$ = 0.0001). Consequently, a significant negative standard minus deviant difference was elicited in response to pattern changes independent of anterior position (Table 1) when absolute pitch information was available. In contrast, in the transposed condition there are no additive but only interactive effects of stimulus and anteriority (stimulus$_{tra}$: $F(1,16)$ = 1.790, $p$ = .200, $\eta_g^2$ = 0.003; anteriority$_{tra}$: $F(2,32)$ = 0.703 $p_{GG}$ = .432, $\eta_g^2$ = 0.001; stimulus*anteriority$_{tra}$: $F(2,32)$ = 10.990, $p$ < .001, $\eta_g^2$ = 0.001). Post hoc analysis of paired comparisons between transposed deviants and standards at the respective anterior positions (see Table 1) revealed that a significant MMN is elicited only at frontal and frontocentral ($t(16)$ > 1.4; $p_{adj}$ ≤ .033) but not at central electrodes ($t(16)$ = 10.313; $p_{adj}$ = .126). Furthermore, the

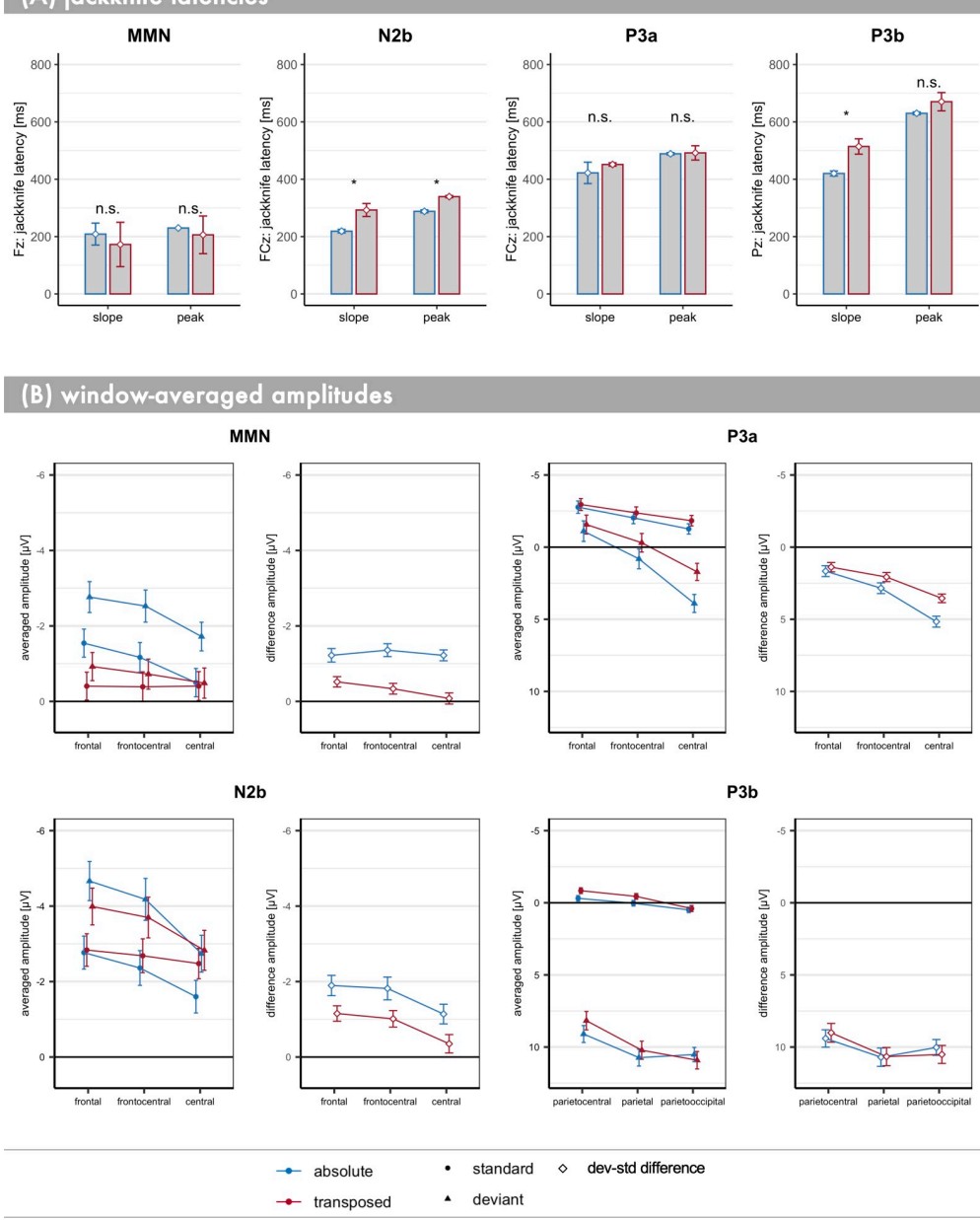

**Fig 4. Error bar plots of averaged ERP component latencies and amplitudes.** Condition is coded by colour (blue: absolute, red: transposed). Error bars represent ± 1 SEM. For exact values please refer to Table 1. (A) Jackknife latency estimates of slope (60% of the local peak) and peak (100% of local peak) in each condition for each ERP component respectively. Note that different electrodes were used for the components (y-axis label). Significant differences ($p_{adj} \leq$ .05) are marked with an asterisk. (B) For each component window-averaged amplitudes are depicted both on the stimulus as well as the deviant-standard difference level from anterior to posterior electrodes along the midline, though averaged across the lateral dimension within a 3x3 electrode array.

amplitude of MMN is statistically significantly larger in the absolute than in the transposed condition at all anterior positions ($t(16) \leq$ -2.034; $p_{adj} \leq$ .033), increasing from medium to large effect size from frontal towards central electrodes (see Table 1). Taken together, MMN in the absolute compared to the transposed condition receives some contribution from more central sources.

**N2b.** Topographies of the averaged difference amplitude values within the respective N2b time windows of each condition (Fig 3B) show that the negative difference between deviants and standards is mainly distributed at frontocentral electrodes, though the field is broader and slightly more right-hemispheric in the absolute compared to the transposed condition.

*Latencies*. Jackknife latencies were extracted within a search window between 180 ms and 500 ms. The second negative difference between deviants and standards, N2b, on average peaks 51 ms ($SE$ = 7 ms) later in the transposed than in the absolute condition, $t_{adj}(16) = 7.020$, $p_{adj} < .001$, $d = 1.703$. Actually, the temporal lag between the conditions which is already present at the time point at which 60% of the respective peak amplitude is reached (N2b slope difference: $M$ = 74 ms; $SE$ = 22 ms), $t_{adj}(16) = 3.287$, $p_{adj} \leq .008$, $d = 0.797$, indicates that there is a positive temporal shift (i.e., increased latency) of the whole component in the absence of absolute pitch information.

*Amplitudes*. Within the N2b time range, activity shows a typical N2 topography, as negativity increases from central towards frontal electrodes, main effect of anteriority: $F(2,32) = 11.583$, $p_{GG} < .001$, $\eta_g^2 = 0.020$. This distribution of activity slightly differs between conditions, condition*anteriority, $F(2,32) = 24.211$, $p_{GG} < .001$, $\eta_g^2 = 0.002$. The anterior increase of negativity is of bigger effect size in the absolute compared to the transposed condition (anteriority$_{abs}$: $F(2,32) = 17.674$, $p_{GG} < .001$, $\eta_g^2 = 0.038$; anteriority$_{tra}$: $F(2,32) = 5.389$, $p_{GG} = .024$, $\eta_g^2 = 0.010$).

Activity within the N2b time range in response to deviants is generally more negative compared to standards, main effect of stimulus: $F(1,16) = 11.720$, $p = .003$, $\eta_g^2 = 0.032$. This deviant minus standard negative difference is more pronounced at frontal than at central electrodes, stimulus*anteriority: $F(2,32) = 16.332$, $p_{GG} < .001$, $\eta_g^2 = 0.003$.

The combination of all these effects translates, as can be seen in Table 1, into the elicitation of a significant N2b regardless of pitch information. Although the negative difference between deviants and standards seems distributed more broadly in the absolute than in the transposed pitch information context (Fig 3B), there are no significant differences in N2b amplitudes between conditions within the 3x3 electrode array (condition*stimulus: $F(1,16) = 3.112$, $p_{GG} = .097$, $\eta_g^2 = 0.003$; condition*stimulus*anteriority: $F(2,32) = 0.056$, $p_{GG} = .945$, $\eta_g^2 < 0.001$).

**P3a.** The grand averaged difference waveforms show that after the N2b peak the deviants are less negative than the standards at anterior electrodes, resulting in a positive difference. Topographies of averaged amplitudes show that especially in the absolute condition a strong posterior positive component (P3b) overlaps with P3a activity, also visible in increased P3a amplitudes from frontal to central in the absolute compared to the transposed condition.

*Latencies*. Jackknife latencies were estimated within a search window from 250 ms and 550 ms. The positive difference between deviants and standards (P3a) at FCz succeeding the N2b peak, reaches 60% of its respective final peak approximately 29 ms ($SE$ = 34 ms) later in the transposed compared to the absolute condition. Though, the difference is not significant, P3a slope: $t_{adj}(16) = 0.860$, $p_{adj} = .405$, $d = 0.207$. The 3 ms ($SE$ = 24 ms) peak latency difference between conditions is not significant either, P3a peak: $t_{adj}(16) = 0.130$, $p_{adj} = .564$ $d = 0.030$. This indicates that P3a was elicited at a relatively similar time in both conditions.

*Amplitudes*. Overall, activity within the time window selected for P3a amplitude extraction both standards and deviants show a decrease in negativity from frontal towards central electrodes, main effect of anteriority: $F(2,32) = 31.735$, $p_{GG} < .001$, $\eta_g^2 = 0.081$. There is no significant main effect of condition alone, $F(1,16) = 3.494$, $p < .080$, $\eta_g^2 = 0.011$. However, amplitudes differ between conditions as a function of anteriority, condition*anteriority interaction: $F(2,32) = 9.705$, $p_{GG} = .005$, $\eta_g^2 = 0.003$. While averaged across stimulus types amplitudes significantly decrease in negativity from frontal to central in both conditions (anteriority$_{abs}$: $F(2,32) = $

40.792, $p_{GG} < .001$, $\eta_g{}^2 = 0.135$; anteriority$_{tra}$: $F(2,32) = 18.080$, $p_{GG} < .001$, $\eta_g{}^2 = 0.068$), this trend is of bigger effect size in the absolute compared to the transposed condition.

Averaged across conditions amplitudes within the P3a time range are significantly more positive in response to deviants than to standards, main effect of stimulus: $F(1,16) = 15.607$, $p = .001$, $\eta_g{}^2 = 0.120$. This deviant-standard difference increases from frontal towards central electrodes, interaction stimulus*anteriority: $F(2,32) = 21.738$, $p_{GG} < .001$, $\eta_g{}^2 = 0.024$. Furthermore, the former effect is modulated by the factor condition, condition*stimulus*anteriority interaction: $F(2,32) = 8.529$, $p_{GG} < .05$, $\eta_g{}^2 = 0.001$ and also a slight interaction between stimulus, anteriority and laterality ($F(4,64) = 3.878$, $p_{GG} = .006$, $\eta_g{}^2 = 0.001$). The positive deviant-standard difference increases more strongly in the absolute compared to the transposed condition from frontal towards central electrodes (see Table 1). This converges with the already described posterior positive component (P3b), building up later in the transposed compared to the absolute condition, resulting in a stronger overlap at the time of the P3a peak in the latter condition. Post hoc analysis of paired comparisons of the difference amplitudes between conditions at the respective anterior positions did not reach significance though, $t(16) < 1.8$; $p_{adj} > .060$; (Table 1). Thus, there is no evidence that the P3a elicited in response to task relevant true pattern changes is reduced in the absence of absolute pitch cues.

**P3b.** Approximately at the time of the auditory pattern offset a strong and broad posterior positivity begins to develop in the grand averaged response to deviants relative to standards, though visually later in the transposed relative to the absolute condition. In both conditions the positive difference steadily rises, the temporal delay in the transposed condition already visible in the slope. The broad peak is visually in temporal proximity to the time point of the average behavioural response. The decline after the peak visually maintains the delay between conditions, indicating a temporal shift of the whole component in the absence of absolute pitch information.

*Latencies*. Jackknife latencies were extracted from a search window between 300 ms and 800 ms. At the time point at which 60% of the final difference peak is reached, there is a significant temporal delay of 94 ms ($SE = 27$ ms) between the transposed and the absolute condition, P3b slope$_{tra\ vs\ abs}$: $t_{adj}(16) = 3.533$, $p_{adj} \leq .007$ $d = 0.857$. The temporal delay when the maximal positive difference between deviants and standards is reached amounts to 40 ms ($SE = 30$ ms), although not significant, P3b peak$_{tra\ vs\ abs}$: $t_{adj}(16) = 1.330$, $p_{adj} = .253$ $d = 0.323$. This is likely due to the broadness of the P3b peak, which makes peak latency estimation difficult.

*Amplitudes*. On average the amplitudes within the P3b time window significantly increase in positivity towards posterior, main effect of posteriority: $F(2,32) = 23.046$, $p_{GG} < .001$, $\eta_g{}^2 = 0.044$. There is no significant main effect of condition on activity within the P3b window, $F(1,16) = 0.455$, $p = .510$, $\eta_g{}^2 = 0.003$. However, the distribution of the posterior activity is modulated by the condition, condition*posteriority interaction: $F(2,32) = 27.379$, $p < .001$, $\eta_g{}^2 = 0.004$. Overall, deviants are significantly more positive than standards, main effect of stimulus: $F(1,16) = 114.369$, $p < .001$, $\eta_g{}^2 = 0.735$. The condition*stimulus interaction is not significant, $F(1,16) = 0.0003$, $p = .986$, $\eta_g{}^2 < 0.0001$. However, the distribution of activity along the midline within the P3b time window differs between standards and deviants, stimulus*posteriority: $F(2,32) = 14.774$, $p_{GG} < .001$, $\eta_g{}^2 = 0.010$, which in turn is modulated by experimental condition, condition*stimulus*posteriority, $F(2,32) = 3.653$, $p = .004$, $\eta_g{}^2 = 0.001$. This means that P3b activity slightly differs between conditions with regard to its distribution along the midline (see also Fig 4B, bottom panel).

However, post hoc analysis further confirmed that a robust and large effect sized P3b (Table 1) is elicited in response to deviant patterns at all posterior positions ($t(16) > 2.7$; $p_{adj} < .004$), and there are no significant P3b amplitude differences as a function of pitch information ($|t(16)| < 0.6$; $p_{adj} > .239$).

## Discussion

Although absolute spectral features are a dominant aspect in auditory processing and absolute pitch information typically is a salient feature of auditory events [9, 13, 80, 81], a certain tolerance to its variability is required, even in initial learning situations. In an indirect listening task Bader and colleagues [17] reported a striking divergence between the relatively untinged automatic change detection in the face of absolute pitch variability at the level of MMN on the one hand, and a prominent decreased further evaluation processing at the level of P3a. While MMN elicitation offers strong indication of sensory learning without absolute pitch cues, the reported P3a amplitude decrease and latency increase [17] are less clear in their meaning. They might reflect a difference in stimulus discriminability [17, 82], computational complexity [83, 84], bottom-up relevance [22, 27, 38, 39], or a combination of these factors depending on whether or not relative pitch distances have to be represented when absolute pitch alone is not sufficient to inform pattern discrimination.

Thus, the aim of the experiment on hand was to assess how pattern processing as a function of pitch reflects on the electrophysiological level in a setting in which the occurrence of new patterns is top-down relevant. Within a roving standard paradigm, randomly generated six-tone patterns were either repeated identically within a stimulus train, carrying absolute pitch information about the pattern, or shifted in pitch (transposed) between repetitions, so only relative pitch information was available to extract the pattern identity. Importantly, participants were asked to indicate whenever they detected a true pattern change, that is to ignore transpositions in the relative pitch context.

### Behavioural performance

As hypothesised and congruent with the findings by Bader and colleagues [17] true pattern changes were detected well above chance regardless of pitch information. This clearly shows that, when explicitly relevant, invariant patterns can be extracted and discriminated from each other despite variable absolute pitch. This is in line with other findings that non-musicians adeptly recognise transposed melodies [7–9, 30, 31, 85]. There were some inter-individual differences, but the largest portion of performance variance was explained by whether absolute or relative pitch cues had to be extracted. Sensitivity to true pattern changes was significantly reduced when relative pitch had to be extracted from transposed pattern sequences. The average response to correctly identified true pattern changes was 67 ms slower when pattern discrimination had to rely on relative pitch extraction. Thus, although true pattern changes were explicitly relevant and transpositions explicitly irrelevant, there is a clear performance advantage of absolute over relative pitch information in pattern discrimination. Similar advantageous effects of absolute pitch (same key) over relative pitch (different key) in melody recognition tasks have been reported [7–9, 30, 31, 85]. This indicates that relevance alone does not provide a sufficient explanation for the P3a effects reported by Bader et al. [17], and implies that there are more fundamental differences in the processing of absolute and relative patterns. Relative pitch related decreased accuracy and increased response times together suggest that there are differences in computational complexity with regard to increased processing time but perhaps also the reliance on additional or even different processes altogether.

### EEG

As hypothesised and similar to other studies using active discrimination paradigms [38, 41], the ERP in response to true pattern changes compared to true pattern repetitions (difference ERP) was characterised by the components MMN, N2b and the P300 complex (P3a and P3b). In comparison to the data reported by Bader et al. [17] MMN and P3a are present in both

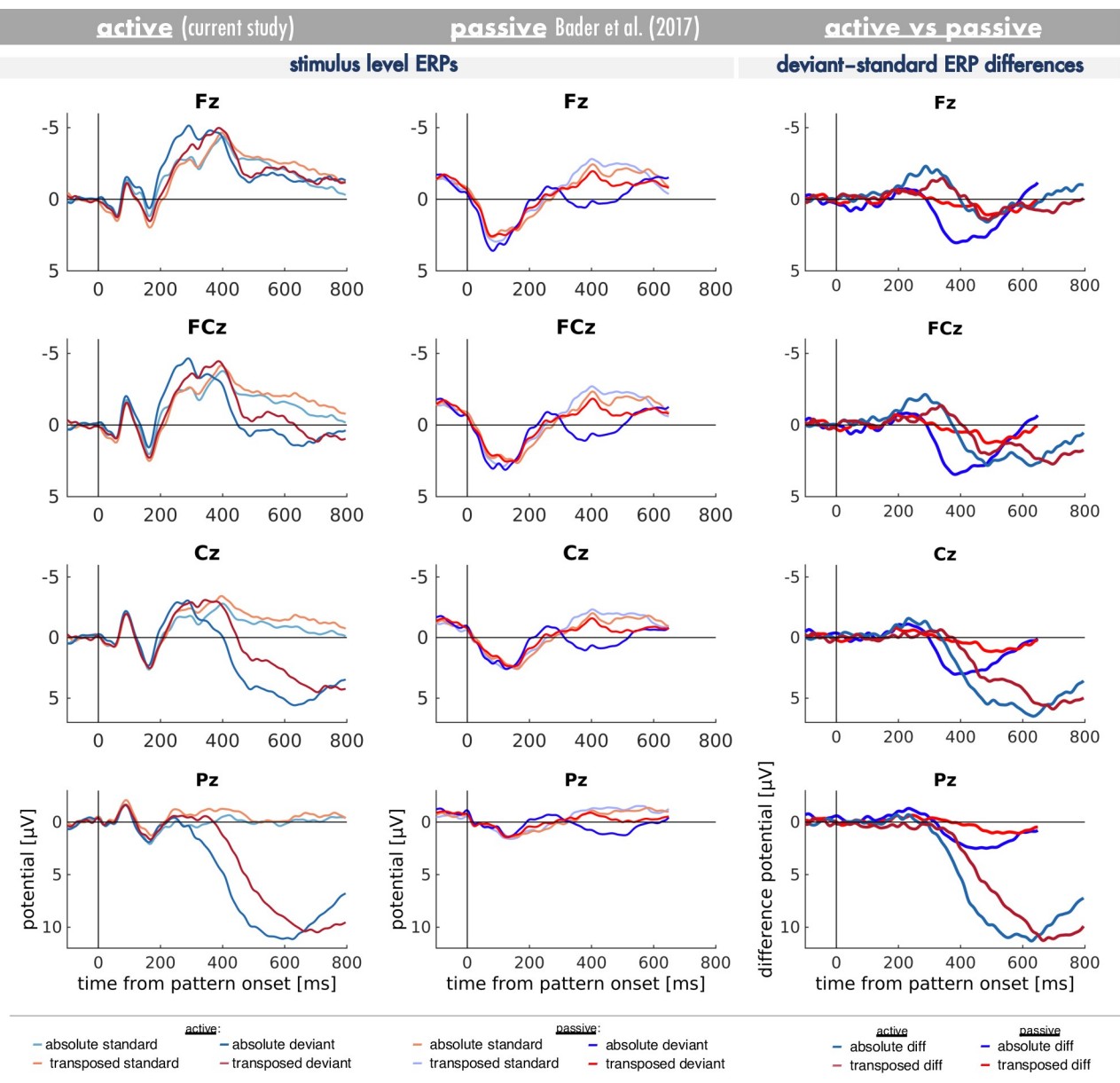

**Fig 5. Comparison of ERPs between active and passive listening setting.** Grand-averaged stimulus-level (standard and deviant pattern) ERP waves as a function of pitch information context (blue: absolute, red: transposed) are depicted in the active listening paradigm of the current study (*left panel*) and the passive listening paradigm by Bader et al. [17] (*middle panel*). Grand-averaged deviant minus standard difference ERPs are compared between active and passive listening setting (*right panel*). Please note that the stimulus-onset-asynchrony was shorter in the passive listening study (650 ms) than in the active listening study (1100 ms) and that train lengths differed slightly as well.

paradigms, whereas N2b and P3b were only observed in the active listening task of the current study but not in the passive listening data (please refer to Fig 5 contrasting the results in active and passive settings).

**MMN.** Activity within the time frame of MMN was significantly larger in the absolute compared to the relative pitch context. While this negative difference could reflect a true MMN amplitude difference, it could also be explained by differential N2b overlap between conditions [18, 19, 42]. In the following we will discuss both accounts.

A potential true MMN amplitude difference could indicate differences in regularity representation strength, as a the stability of auditory environment has been inversely linked to model precision [86]. To elaborate, it is suggested that precision is higher the more stable an auditory regularity (low variability), and when deviation occurs from the expectation under a high-confidence scenario (higher precision), it results in larger MMN [86]. Within such an interpretational framework differences in MMN amplitude would reflect decreased model precision as a result of high environmental variability. However, there is some argument that MMN amplitude is not proportional to deviation magnitude, but that it is rather an all-or-none response [87]. This means, that differences in the average MMN amplitude between conditions would reflect a differential consistency with which MMN was elicited respectively. Detection of a new pattern needs at least two segments in both conditions. However, there might have been some jitter in (perceptual) change onset between conditions, resulting in a temporally more variable transposed MMN, not as well captured by a window averaging approach.

However, in the context of absolute pitch there was only one prominent, quite broadly distributed frontocentral negative deflection with an early shoulder within the typical latency range of MMN but the main peak occurring in the time range of N2b, impeding a robust estimation of MMN amplitude and latency. Thus, a likely explanation for the observed amplitude differences as a function of pitch context is a differing degree of temporal and spatial superposition of MMN by N2b [18, 19, 42], especially considering the more central contributions to the negative deflection in the MMN window in the absolute compared to the relative pitch context. In fact, MMN and N2b are often dissociated by the somewhat more central distribution of the latter component compared to the former, and typically also by polarity inversion at mastoids for MMN but not N2b [18, 19, 42, 88, 89]. However, the processing of complex patterns generally tends to elicit a more central MMN topography [17] than prototypical classic oddball stimuli, complicating the first approach. Furthermore, our observations seem similar to Bader et al. [17] that such patterns appear to also lack a pronounced and clear inverted peak at the mastoids.

One alternative approach to disambiguate MMN and N2b component overlap is to contrast a passive and an active oddball condition [90–92]. The direct comparison with the passive listening paradigm by Bader et al. [17] (see Fig 5) suggests that the morphology of MMN in response to deviant patterns in the transposed condition in our active paradigm is highly concordant with the passively elicited MMN–both with regard to latency and amplitude. In the absolute condition, the initial portion of the negativity shows a similar concordance, with a divergence starting at around 210 ms (that is 160 ms after deviance onset), which is well in line with typical slope latencies of N2b [46, 47, 89, 93]. Thus, MMN shows a similar pattern in both studies, whereas the relatively later negativity (N2b) is only observable in the active but not the passive listening condition.

Particularly in the context of relative pitch, MMN was distinctly identifiable from N2b. It peaked at approximately 206 ms after stimulus onset, considering the 50 ms delay due to the fixed first segment in both experimental conditions, around 150 ms after the actual change in pattern identity started to occur. That is well within the typical MMN peak range [18, 19] and concords with other MMN-studies on higher-order regularities [15, 41, 94–96]. These results are in line with Bader et al. [17] that a significant MMN is elicited in response to true pattern changes even when relative pitch information has to be extracted because absolute pitch varies. In accordance with common MMN interpretation [16, 97] these results support the inference by Bader et al. [17] that relative pitch information is sufficient for the formation of a sensory memory trace of a regular auditory pattern and its discrimination from other relative patterns at least on the sensory level.

**N2b.** N2b was elicited in response to true pattern changes in both absolute and relative pitch context. There was a descriptive, though not significant, attenuation of the amplitude for relative compared to absolute true pattern changes. N2b amplitude often is interpreted as an indicator of the strength of a voluntarily held stimulus template [45, 47], i.e. in this case how well the spectrotemporal pattern is represented based either on absolute or relative pitch information. The lack of evidence for an amplitude modulation might mean that a pattern template is established equally well in both instances. However, N2b amplitude is also taken to reflect the allocation of attentional resources. If relative compared to absolute patterns require more attentional resources at the level of N2b, that might have cancelled out any effects of representation strength. Notably, findings on N2b amplitude modulations are generally rather inconsistent: some studies report a more negative [98], others a less negative N2b amplitude [99] with increasing discrimination difficulty. Similarly, with progressing age both increases [38, 100, 101] as well as decreases [102] were observed. No evidence for N2b amplitude modulations was reported for age [103] and medication in ADHD patients [104]. Ultimately, N2b amplitude might simply be slightly smaller because it temporally somewhat overlapped more with the stimulus offset response [14] and the P3a in the relative compared to the absolute pitch context.

More noteworthy might be the significant temporal shift of the whole N2b component in the absence of pitch information in the order of magnitude of the respective delay on the level of reaction times. N2b latency has been observed to increase with discrimination demands [44, 46, 98, 105], and with age-related decline of cognitive resources [38, 100, 103]. Interestingly, N2b latency in children diagnosed with ADHD has been reported to be abnormally short compared to neurotypicals, and to not only be normalised (i.e., increased) under medication with methylphenidate but also, to concur with improved behavioural accuracy [104]. Taken together, it seems that N2b latency not merely indexes the speed but also the complexity, and perhaps accuracy, of the processing of deviating and, or target stimulus characteristics. In these terms, the increase in latency in the relative compared with the absolute pitch context, might well reflect generally increased computational complexity when a pattern is defined by relative rather than absolute pitch. As Ritter et al. (1992) pointed out that the interpretation of N2b largely relies "on the experimental circumstances and the strategies used by the subjects", one could argue even further: N2b latency differences as a function of pitch might not just reflect differences in mere processing time between absolute and relative patterns, but might actually indicate wholly different kind of processes operating at the stage of stimulus comparison or categorisation respectively [100, 103].

**P3a.** In a passive listening situation Bader et al. [17] reported substantial P3a impoverishments for relative compared to absolute pitch context in the form of a delayed peak (approximately 130 ms) and diminished amplitude. One possible explanation for these effects is the question of relevance [22, 27, 38, 39]. Whereas a new pattern in a context of identical (absolute) pattern repetitions constitutes a clear deviation to the preceding stimulation, the occurrence of a new pattern in a relative pitch code context must not necessarily be inherently more relevant to the brain than if it were a transposition of the preceding pattern–in both cases absolute pitch information has changed. Thus, at least the P3a amplitude modulation by Bader et al [17] might have reflected decreased allocation of attentional resources to the true pattern changes.

The current study aimed to assess the validity of that argument, by making the true pattern changes but not the transpositions task relevant. There was no evidence that the P3a elicited in response to task relevant true pattern changes is reduced and none that P3a latency was affected in the absence of absolute pitch cues. Whatever differences exist in relative compared to absolute pattern processing in general, there was no evidence in the current study to suggest

that when true pattern changes are explicitly relevant, that they manifest in the processes operating at the stage of P3a. In active listening the processes at P3a level seem to occur relatively independent of whether absolute or relative pitch information defines the patterns.

However, caution should taken when pondering whether this is evidence that top-down task-relevance of true pattern changes compensates for lack of bottom-up salience [24] in passive listening. It is not entirely clear yet whether P3a represent the same underlying processes in passive and active listening situations [24]. Actually, in the direct comparison of passive and active listening task (Fig 5), it appears that P3a elicited by true pattern changes might be separable into two distinct subcomponents [106–110]–namely an early and a late P3a. When visually comparing both studies a potential late P3a was elicited of similar latency and amplitude in both studies irrespective of pitch information. However, a potential early P3a was elicited only in response to absolute true pattern changes in passive listening. Thus, this early P3a activity could be a candidate to further elucidate differential pattern processing as a function of pitch in future studies. It might well be the case that processing of relative patterns depends to some degree on the investment of direct attentional resources, while absolute patterns can be processed relatively independent from attention. Then again, in passive listening neither transpositions nor true pattern changes hold behaviourally relevant information for the listener. Even though the human auditory system might be able to discriminate relative patterns from each other just as well absolute patterns, it might be actually beneficial to ignore the constantly changing auditory environment in a relative pitch context. Whether learned relevance could compensate in the passive listening needs to be adressed in future studies including a passive listening task following an active learning phase.

**P3b.** As expected a prominent posterior P3b was elicited when a target (i.e. a true pattern change) was correctly identified [21–24, 48]. There was no evidence for an attenuation of P3b amplitude but a clear and pronounced delay in the absence of absolute pitch information.

Within the traditional framework [51, 52] the lack of evidence for modulation of P3b amplitude by pitch information implies that during the revision, or updating of the current mental model in response to the detection of a true pattern change, attentional resources are equally available [22, 38]. A more recent account suggests that P3b rather reflects reactivation of stimulus-response links, and that P3b amplitude increases as a function of response infrequency [111]. The absence of P3b amplitude modulation in the current study would thus be explained by the fact that true pattern changes occurred equiprobably in both absolute and relative pitch context, resulting in comparable response frequencies.

Within both contexts, the increase in P3b latency would signify increased stimulus evaluation time [22, 48, 112]. Indeed, the observed delay was already present at the N2b level, and did not notably increase at the level of P3b. That P3b peaked after the correct behavioural response was made [24], implies that it reflects post-identification stimulus categorisation, response selection and execution processes [24, 48, 49, 55, 56, 111] which are not vulnerable to the absence of pitch information.

## Conclusion

To conclude, even with specific instruction there is a clear advantage of absolute over relative pitch information in the active discrimination of unfamiliar melodic patterns. When relative pitch has to inform pattern learning and discrimination, the most notable electrophysiologic correlates are increased latencies at the level of N2b and P3b. Interestingly, the response delay of approximately 70 ms on the behavioural level, already fully manifests as early as at the level of N2b and is merely propagated to the level of P3b. In contrast, MMN and P3a were elicited regardless of pitch information to inform pattern discrimination. In sum, these findings

strongly suggest that, rather than a break-down of the auditory change detection process, or later working memory or response related processes, target selection is at the root of the deterioration in behavioural performance. Specifically, relative compared to absolute pitch processing during active pattern learning either or both differs with regard to increased processing time but perhaps also the reliance on additional or even different computational processes altogether.

## Supporting information

**S1 File. Example of absolute pattern sequence depicted in Fig 1.**
(WAV)

**S2 File. Example of transposed pattern sequence depicted in Fig 1.**
(WAV)

## Author Contributions

**Conceptualization:** Nina Coy, Maria Bader, Erich Schröger, Sabine Grimm.

**Data curation:** Nina Coy, Maria Bader, Sabine Grimm.

**Formal analysis:** Nina Coy, Maria Bader, Sabine Grimm.

**Funding acquisition:** Sabine Grimm.

**Investigation:** Nina Coy, Sabine Grimm.

**Methodology:** Nina Coy, Erich Schröger, Sabine Grimm.

**Project administration:** Maria Bader, Sabine Grimm.

**Resources:** Erich Schröger, Sabine Grimm.

**Software:** Nina Coy, Sabine Grimm.

**Supervision:** Erich Schröger, Sabine Grimm.

**Validation:** Nina Coy, Maria Bader, Erich Schröger, Sabine Grimm.

**Visualization:** Nina Coy, Sabine Grimm.

**Writing – original draft:** Nina Coy.

**Writing – review & editing:** Nina Coy, Maria Bader, Erich Schröger, Sabine Grimm.

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
