## [Decision Letter · Decision Letter 0]

18 Nov 2020

PONE-D-20-25390

Change detection of auditory tonal patterns defined by absolute versus relative pitch information. A combined behavioural and EEG study.

PLOS ONE

Dear Dr. Coy,

Thank you for submitting your manuscript to PLOS ONE. After careful consideration, we feel that it has merit but does not fully meet PLOS ONE’s publication criteria as it currently stands. Therefore, we invite you to submit a revised version of the manuscript that addresses the points raised during the review process.

Note both reviewers have major concerns about the data analysis. It is highly recommended to include the additional source space analysis or PCA analysis in the revision. Such analyses may be critical to support the conclusion in the present study. 

We look forward to receiving your revised manuscript.

Kind regards,

Qian-Jie Fu, Ph.D.

Academic Editor

PLOS ONE

Journal Requirements:

Reviewers' comments:

Reviewer's Responses to Questions

**Comments to the Author**

1. Is the manuscript technically sound, and do the data support the conclusions?

Reviewer #1: Partly

Reviewer #2: Partly

2. Has the statistical analysis been performed appropriately and rigorously? 

Reviewer #1: Yes

Reviewer #2: No

3. Have the authors made all data underlying the findings in their manuscript fully available?

Reviewer #1: Yes

Reviewer #2: Yes

4. Is the manuscript presented in an intelligible fashion and written in standard English?

Reviewer #1: Yes

Reviewer #2: Yes

5. Review Comments to the Author

Reviewer #1: This study describes the behavioural and spatiotemporal EEG dynamics associated with detecting melodic oddball deviants that differ in either absolute pitch or relative pitch. A primary purpose was to confirm that complex auditory representations are “automatically” coded in memory even in the absence of absolute pitch information. A secondary purpose was to describe any behavioural (RT) and spatio-temporal EEG differences in so-called “automatic” (as indexed by the MMN and maybe the P3a) and “intentional” (as indexed by the N2b and P3b) processes involved in the active detection of auditory irregularities defined by either absolute pitch deviants or relative pitch deviants. The main results show that while MMN and P3a latencies are similar when detecting absolute and relative pitch deviants, N2b and P3b latencies are longer to relative pitch deviants compared to absolute pitch deviants. This suggests that complex auditory representations may be coded automatically even in the absence of absolute pitch information, while intentional processes are slower to respond to auditory representations in the absence of absolute pitch information.

In general I think this work is well done; the goals are properly-justified, the study design was well thought-out and executed, analyses were mostly clear and properly-justified (but see major revisions), and the results provide interesting insight into the temporal dissociation between automatic coding of auditory representations and intentional higher-order processing (attention, stimulus matching) (but again, see major revisions). I have one major concern and a few minor concerns. I am, therefore, requesting accept with Major Revisions, but I have every confidence the major revision will be addressed.

Major Revisions

Please explain in more detail how you determined that there is, in fact, both an MMN and an N2b in the absolute pitch condition. While I’m sympathetic to the argument that “should” be both components present in the absolute condition, and that the difference wave in Figure 3 represents a likely spatiotemporal overlap between the two conditions, I am, however, concerned that the MMN and N2b scalp maps look very similar. My understanding is that the N2b, unlike the MMN, does not typically have an inversion at temporal sites (e.g. Näätänen & Gaillard, 1983; Sussman et al., 2002 see Fig 2), but the N2b scalp map shown in Figure 3 appears to have just as much source located around temporal/posterior sites as does the MMN scalp map. Since one of the main conclusions of the study is that automatic coding of auditory patterns occurs at a similar timecourse regardless of whether absolute pitch information is available (as indexed by the latency of the MMN), but that intentional processes are slower to detect relevant pitch deviants (as indexed by the latency of the N2b as well as the P3b), it’s vital to your argument that you make clear that both the MMN and N2b were, in fact, generated in each condition.

References:

Näätänen, R., & Gaillard, A. W. K. (1983). 5 The Orienting Reflex and the N2 Deflection of the Event-Related Potential (ERP). In A. W. K. Gaillard & W. Ritter (Eds.), Advances in Psychology (Vol. 10, pp. 119–141). North-Holland. https://doi.org/10.1016/S0166-4115(08)62036-1

Sussman, E., Winkler, I., Huotilainen, M., Ritter, W., & Näätänen, R. (2002). Top-down effects can modify the initially stimulus-driven auditory organization. Cognitive Brain Research, 13(3), 393–405. https://doi.org/10.1016/S0926-6410(01)00131-8

Similarly, please make clearer how you determined the latency of the N2b given the overlap between MMN and N2b. You explain that the MMN latency was defined as the point at which the initial negative deflection reached 60% of the peak amplitude, but it’s not clear how you determined the latency for the N2b. Which slope and which peak amplitude did you use to calculate the latency of the N2b in the absolute condition?

Finally, please make clearer from which channels did you extract the data for your statistical comparisons. From Table 1 it looks like you picked Fz for the MMN and FCz for the N2b. Why did you choose these channels? The table also seems to show that peak difference wave amplitude for the transposed condition, for which there is potentially a clear MMN, was somewhere in the FC region (this is also confusing, see minor revisions), while the N2b in that same condition peaked somewhere in the F region. There are similar discrepancies for P3a and P3b subcomponents.

Minor Revisions

General note: please make sure all your variables are italicized (namely d’s and η^2)

Methods section: if it’s possible and not too much trouble, I’d appreciate a rough estimation of when the deviants actually deviated from the preceding standards (i.e. which note within the deviant stimulus was unexpected relative to the previous standard). I understand difference waves weren’t calculated this way (deviant – immediately preceding standard), but I’d like to know approximately when the average deviation occurred. You state that in the absolute condition the first note was always fixed at 400 Hz, and then you imply that the second note consistently deviated from the previous standard, but I’m not sure if I understood you correctly. This will help provide some context for the ERP latencies you’re reporting, since presumably the latencies (particularly of the early components) reflect about when the auditory and attentional systems detected a violation of the ongoing auditory regularity. If that varied a lot deviant by deviant, and if one condition varied more than the other, then the meaning of the latency differences between conditions becomes harder to interpret. At the very least I think you should provide a range of which notes could have represented deviations from the preceding standards, and perhaps address this potential issue in the discussion.

Methods section: how many different tone patterns were there? Were they randomly generated or predefined?

Results section: please remind the reader of the timeframes you’re using to define each subcomponent in their subsequent “latency” section of the results

Table 1: Why are some of the channel labels in EP ambiguous (i.e. F instead of Fz or F1)? Why not state exactly which channels each row is referring to? Do the rows represent averaged data from multiple channels? If so which channels? Please clarify this in the caption of the table.

Lines 59-60: “…facilitating the recognition of auditory objects despite large variation in spectral features”. This statement should have a reference.

Line 69: please supply more references that speak to the “rich body of research”

Line 88: “statistically robustly”. I assume you mean that there was a strong positive correlation between deviant ERP amplitude and the number of preceding standard stimuli? If that’s the case then I don’t think you need to include “statistically robustly”, as it reads as a subjective evaluation of the correlation. It’s enough to say that the amplitude increased as a function of the number of preceding stimuli.

Lines 107-109: Please support this sentence with a reference.

Line 564: “the reported P3a amplitude decrease and latency increase…”. Fig 4a shows that P3a latencies were not different. Please clarify.

Lines 565-568: Please include references for how the P3a is associated with any of the listed stimulus features/cognitive demands

Line 586: “notably and significantly reduced…”. Please refrain from using subjective evaluations (notably) when statistical comparisons suffice (statistically).

Reviewer #2: This paper is a follow up of a previous paper by Bader et al, in which, during a passive listening paradigm, no difference in MMN was found in response to pattern violations with absolute pitch or relative pitch changes. This paper did found decreased amplitude and increased latency of P3a, however, in the relative pitch compared to absolute pitch condition. The present paper conducts a similar study, but now in an active paradigm in which participants are asked to indicate when a repeating stimulus pattern changes, in conditions involving absolute versus relative pitch cues. The basic findings were that now P3a was similar in amplitude and latency across for the relative and absolute pitch conditions. And N2b and P3a were both delayed in the relative pitch compared to absolute pitch conditions.

The paper is interesting in attempting to dissect the stages of processing in order to determine where relative and absolute pitch information in patterns are processed similarly and differently. However I have some concerns.

1. A general difficulty with the approach is that there is not wide consensus in the literature as to exactly what these 4 components represent, leaving the paper with considerable speculation as to the meaning of the results.

2. The literature review on MMN, P3a and relative pitch is not very complete. For example, it is stated “Since then, a rich body of research has emerged, enabling a better understanding of the processing of complex sounds”, with no references given. For example, the work of Trainor and colleagues showed MMN and P3a responses for relative pitch information in non-musicians (e.g., 2002, J Cog Neuro) and infants (e.g., He et al., 2009 Eur J Neurosci). As well, musical training may significantly modify these brain responses to relative pitch information (e.g., Fujioka et al., 2004; J Cog Neuro), as may exposure to a tonal language, so the generalizability of the results should be considered in the discussion.

2. Methodologically, I think several major issues need to be addressed as follows.

2a. Stimuli: the stimulus patterns are very brief with the tones comprising them only 50 ms. The whole patterns are only 300 ms. Given that the temporal window of integration is 150 – 200 ms, it is not clear that these are perceived as patterns in the sense of a sequence of events. Rather they are likely processed as single units.

2b. The 4 components of the ERPs (MMN, N2b, P3, P3b) are difficult to separate and extract using waveforms from surface electrodes. Especially given the different topographies of the components that overlap, and the differences between conditions, the analyses should be done after source localization of each components has been done. Right now, not only can the components not be cleanly separated, but the ANOVA statistical approach is complicated by many factors and complex interactions. This makes the paper very difficult to read and raises the problem of many statistical tests. All of this would be much simpler and cleaner if done in source space. Furthermore, if I understand correctly, single electodes were analyzed, thus ignoring the rich information available from having collected 64 channels of data. Again sources space analysis would include all information across the scalp. Alternatively some kind of PCA analysis might also be able to better separate and characterize the components by utilizing all of the data from all electrodes over time.

2c. The jackknife approach needs more explanation. It is not clear if it was applied across participants or whether an individual latency was calculated for each participant.

2d. The use of when 60% and 100% of the peak of some of the components was not clear and seems arbitrary in order to make arguments that some effects of significant at 60% even though not at 100%. This needs to be justified or should be removed.

2e. There is a fundamental problem in separating and distinguishing the MMN and N2b components in the absolute pitch condition. Thus, without doing some kind of source localization, both the MMN and N2b results are unclear.

3. Results/interpretation

3a. MMN was actually found to be a bit larger in absolute condition compared to the relative condition, which is actually not in agreement with Bader et al. However, this is inconclusive as, with the current analysis methods, it was not really possible to separate MMN and N2b. The authors argue that because of this MMN is not really different across conditions, but I think this remains unknown.

3b. With respect to N2b it was found to be later in the relative than absolute pitch condition. However, this is not completely convincing as the MMN and N2b could not be completely separated. This delayed N2b for relative pitch is interpreted as increased discrimination demands, but it seems that other interpretations might be possible.

3c. The amplitude of the N2b was “descriptively” smaller in amplitude for the relative pitch case, but not statistically different. A somewhat convoluted argument is made that N2b might reflect both “representational strength” and “attention” and that the lack of significant amplitude difference might reflect some combination of reduced “representational strength” and increases “attention in the relative pitch case. This seems overly speculative.

3d. No amplitude or latency differences were found for P3a for relative and absolute conditions. This is consistent with the idea that top-down processes (attention?) compensate for any increased difficult of the relative pitch condition. But the authors go on to state the absolute and relatively pitch information might just be processed “differently”. I couldn’t follow this argument.

3e. The P3b result was quite clear – no amplitude different but a longer latency for relative than absolute condition.

3f. From the results, the N2b and P3b were both delayed in the relative compared to absolute condition. The authors interpret the N2b as representing “target selection” and the P3b as representing change detection, or working memory or response related processing. Therefore, they conclude that target selection is delayed from relative pitch, but that change detection, working memory and response-related processing are not of themselves delayed. This argument of course depends on accurate assessment of N2b latency.

3g. I find it curious that in the conclusions, no mention is made of MMN or P3a. There is no discussion of how these two component relate to N2b and P3b in a stages of processing account. And why, for example, N2b can be delay while the following P3a is not delayed.

In sum, this is an interesting study, but many questions need to be addressed before the results can be interpreted with confidence.

6. PLOS authors have the option to publish the peer review history of their article (what does this mean?). If published, this will include your full peer review and any attached files.

Reviewer #1: No

Reviewer #2: No

---

## [Author Response · Author response to Decision Letter 0]

23 Dec 2020

Dear Reviewers,

We are pleased to resubmit for publication the revised version of the manuscript entitled “Change detection of auditory tonal patterns defined by absolute versus relative pitch information. A combined behavioural and EEG study.” by Nina Coy, Maria Bader, Erich Schröger, and Sabine Grimm. 

We appreciate your constructive comments. We will address the central concern regarding ERP component separation first. Each of your other concerns are addressed in the later sections.

Separation of ERP components, especially MMN and N2b

Reviewer #1: “Please explain in more detail how you determined that there is, in fact, both an MMN and an N2b in the absolute pitch condition. While I’m sympathetic to the argument that “should” be both components present in the absolute condition, and that the difference wave in Figure 3 represents a likely spatiotemporal overlap between the two conditions, I am, however, concerned that the MMN and N2b scalp maps look very similar. My understanding is that the N2b, unlike the MMN, does not typically have an inversion at temporal sites (e.g. Näätänen & Gaillard, 1983; Sussman et al., 2002 see Fig 2), but the N2b scalp map shown in Figure 3 appears to have just as much source located around temporal/posterior sites as does the MMN scalp map. Since one of the main conclusions of the study is that automatic coding of auditory patterns occurs at a similar timecourse regardless of whether absolute pitch information is available (as indexed by the latency of the MMN), but that intentional processes are slower to detect relevant pitch deviants (as indexed by the latency of the N2b as well as the P3b), it’s vital to your argument that you make clear that both the MMN and N2b were, in fact, generated in each condition.”

Reviewer #2: “The 4 components of the ERPs (MMN, N2b, P3, P3b) are difficult to separate and extract using waveforms from surface electrodes. Especially given the different topographies of the components that overlap, and the differences between conditions, the analyses should be done after source localization of each components has been done. Right now, not only can the components not be cleanly separated, but the ANOVA statistical approach is complicated by many factors and complex interactions. This makes the paper very difficult to read and raises the problem of many statistical tests. All of this would be much simpler and cleaner if done in source space. Furthermore, if I understand correctly, single electodes were analyzed, thus ignoring the rich information available from having collected 64 channels of data. Again sources space analysis would include all information across the scalp. Alternatively some kind of PCA analysis might also be able to better separate and characterize the components by utilizing all of the data from all electrodes over time.”

We fully agree with the Editor and Reviewers, that more detailed knowledge about the sources or the underlying component structure would immensely help when analysing (and interpreting) the effects of our experimental manipulations. Our research group has expertise in analysing EEG data using source separation methods such as PCA (e.g., Bonmassar, Widmann & Wetzel, 2020; Widmann, Widmann, Schröger & Wetzel, 2018; Scharf & Nestler, 2018; Scharf & Nestler, 2019) and we pursued several attempts of applying a PCA to the data reported in the manuscript. The typical approach would be to decompose data using a temporal PCA in a combined fashion (using standard and deviant ERPs from both the absolute and the transposed condition together as input). However, such combined approaches will usually not satisfactorily unmix the data unless the systematic condition-related latency differences are sufficiently large (in the range of 100 ms; see, Barry et al., 2016). Please note, that this restriction even holds for very prominent components such as the P3b. Thus, for small or medium latency differences between conditions, PCA often results in a single latent component capturing the time course activated by both conditions and a second latent component capturing the difference between the two latencies. Thus, latency shifts across individuals or conditions can result in “ghost” components resembling the time-derivative of the latency behaviour (Mocks 1986; Dien, 1998).

A separate temporal PCA approach (running the PCA for the absolute and the transposed condition separately, Fig 1) is therefore in our case more appropriate. Yet, while PCA (i.e., finding orthogonal directions) might allow the separation of well-distinguishable components, a high overlap in time can prevent the algorithm from finding a satisfactory solution during unmixing (partly also due to similar reasons as mentioned above). We see this in our results: for the absolute condition, a very broad negative component is found (including both peaks that we interpret as MMN and N2b), for the transposed condition, the PCA reveals two components in the time range of MMN and N2b, one that is again broad and two-peaked and one that only includes the later peak of N2b (Fig. 1 bottom panel). This somewhat points in the right direction, yet it is overall difficult to interpret as we know that the results of PCA are affected by the degree of overlap and since we see in our separate PCAs signs of problematic unmixing (component time courses are partly not monophasic and focal). We run into similar problems when applying a spatial PCA (Fig 2): such an approach is more sensitive to the specific rotation method used and while it is consistently able to distinguish between the parietal topography of P3b and a frontocentral negative component, we only see an inconsistent unmixing of frontal and central parts of the topographies of MMN and N2b (which have a probably even higher overlap in the spatial dimension than in the temporal dimension). Spatial PCA generally suffers from overlap of components due to volume conduction (Dien, 2012), which means that typically many or even most electrodes have non-zero loadings on more than one factor (i.e., cross-loadings). This often results in worse component separation performance compared to temporal PCA.

Figure 1 Overview of principal termporal components (TCs) topography and time course retrieved from a temporal PCA (goemin, epsilon = 0.5) for each condition separately. The TC1 in ech condition is a late positivity with parieto-occipital distribution peaking at 658 ms in the absolute and at 700 ms in the transposed condition (reflecting P3b). TC2 in each condition is a broad, partly double-peaked negative component (with frontocentral distribution peaking at around 390 ms) – likely reflecting the MMN/N2b complex. Bottom panel shows projected TCs whose time course falls into the range of MMN/N2b in comparison to the grand averaged deviant-standard difference waves. In the absolute condition one main component was found (TC2 – explaining 26.4 % of the variance in the data) showing a broad time course of frontocentral negativity (roughly from 150 ms – 550 ms). In the transposed condition, TC2 (explaining 20.5% of the variance in the data) has a similarly broad time course (roughly from 150 ms – 550 ms) and is double peaked. An additional component TC3 explains 3.5 % of the variance and coincides mainly with the second peak of the grand averaged ERP difference wave form (likely reflecting N2b). Please note, that the temporal components are overall rather broad and in several instances biphasic. This likely reflects difficulties in unmixing the data due to high temporal and spatial overlap of MMN and N2b.

Figure 2 Top: Overview of spatial components (SCs) topography and time course retrieved from a combined spatial PCA (varimax rotated). The first SC is a component with a parieto-occipital distribution explaining 55 % of the variance in the data (reflecting P3b). The second SC has a frontocentral distribution and is mainly active in broad time window covering MMN/N2b. The third SC is a rather focussed central component also covering wide portions of the MMN/N2b explaining 8 % of the variance in the data. Bottom: Comparison of grand-average difference waveforms and PC projections for the first three SCs The parieto-occipital component SC1 explains the P3b activity in the data to a high degree. SC2 with a frontocentral distributions is active during the time course of MMN/N2b , but partly also during later time ranges. SC3 with a central distribution contributes to an early portion of the negativity (covering MMN and N2b) in the absolute (left panel), but not so much in the transposed condition (right panel). Again this component also contributes to activity in later time ranges. Please note, that the orthogonal nature of PCA will not easily deal with high spatial overlap between components. Whether the central component SC3 is rather a “ghost” source or reflects a true centrally distributed activation contributing to MMN (since it peaks rather early) is unclear – yet MMN topography shifts to more central/parietal regions are sometimes seen for complex stimulus material.

Therefore, we are still highly hesitant whether results of the PCA reflect the real situation better and add trustful, valuable information to the analysis of ERPs.

To not leave the Reviewers’ justified concerns unaddressed, we finally chose a different approach. In the literature, the classical way of distinguishing MMN and N2b is a direct comparison between a passive and an active oddball condition (see Potts et al., 1998; Wei, Chan & Luo, ; Sussman, 2007). Since we have a very similar dataset from the study of Bader et al. (2017), where subjects listened to the same complex tonal patterns in a passive paradigm (only with a different SOA and a slightly different combination of possible train lengths), we decided to directly compare these conditions graphically. In comparison to the data reported by Bader et al. (2017) MMN and P3a are present in both paradigms, whereas N2b and P3b were only observed in the active listening task of the current study. Specifically, the comparison suggests that the morphology of MMN in response to deviant patterns in the transposed condition in our active paradigm is highly concordant with the passively elicited MMN – both with regard to latency and amplitude. In the absolute condition, the initial portion of the negativity shows a similar concordance, with a divergence in between active and passive starting at around 210 ms (that is 160 ms after deviance onset), which is well in line with typical slope latencies of N2b (Breton et al., 1988; Näätänen et al., 2014; Novak, Ritter, Vaughan, & Wiznitzer, 1990; Sams et al., 1983). Thus, MMN shows a similar pattern in both studies, whereas the relatively later negativity is only observable in the active but not the passive listening condition. 

Additionally, this direct comparison yielded another potentially useful insight: it appears that P3a elicited by true pattern changes might be separable into two distinct subcomponents – namely, an early and a late P3a. When visually comparing both studies a potential late P3a was elicited of similar latency and amplitude in both studies irrespective of pitch information. However, a potential early P3a was elicited only in response to absolute true pattern changes in passive listening. Thus, this early P3a activity could be a candidate to further elucidate differential pattern processing as a function of pitch in future studies.

Figure 3 Grand average waves elicited in response to true pattern changes occurring in an absolute and relative (transposed) pitch context during active (left panel; current submission) and passive listening (middle panel; Bader et al., 2017). The right panel shows the grand averaged deviant-standard difference ERPs reported in both studies. 

Additionally, in order to simplify the statistical analysis, we decided to drop the laterality factor from the ANOVA and collapse the cells accordingly. We have uploaded the revised statistical analysis (Rmd Files) to the OSF repository linked in the manuscript.

References

 Barry, R. J., De Blasio, F. M., Fogarty, J. S., & Karamacoska, D. (2016). ERP Go/NoGo condition effects are better detected with separate PCAs. International Journal of Psychophysiology, 106, 50-64.

 Breton F, Ritter W, Simson R, Vaughan HG. (1988) The N2 component elicited by stimulus matches and multiple targets. Biol Psychol;27:23–44.

 Bonmassar C, Widmann A, Wetzel N. (2020) The impact of novelty and emotion on attention-related neuronal and pupil responses in children. Dev Cogn Neurosci;42:1878–9293. https://doi.org/10.1016/j.dcn.2020.100766.

 Dien, J. (1998). Addressing misallocation of variance in principal components analysis of event-related potentials. Brain topography, 11(1), 43-55. 

 Dien J. (2012) Applying principal components analysis to event-related potentials: A tutorial. Dev Neuropsycholy; 37:497–517.

 Mocks, J. The influence of latency jitter in principal component analysis of event-related potentials. Psychophysiology, 1986, 23(4): 480-484.

 Novak GP, Ritter W, Vaughan HG, Wiznitzer ML. Differentiation of Negative Event-Related Potentials in an Auditory-Discrimination Task. (1990) Electroencephalogr Clin Neurophysiol;75:255–75.

 Näätänen R, Sussman ES, Salisbury D, Shafer VL. Mismatch negativity (MMN) as an index of cognitive dysfunction. (2014) Brain Topogr 2;27:451–66. https://doi.org/10.1007/s10548-014-0374-6.

 Potts, G. F., Dien, J., Hartry-Speiser, A. L., McDougal, L. M., & Tucker, D. M. (1998). Dense sensor array topography of the event-related potential to task-relevant auditory stimuli. Electroencephalography and clinical neurophysiology, 106(5), 444-456.

 Sams M, Alho K, Näätänen R. (1983) Sequential effects on the ERP in discriminating two stimuli. Biol Psychol;17:41–58. https://doi.org/10.1016/0301-0511(83)90065-0.

 Scharf & Nestler, S. (2018) Principles behind variance misallocation in temporal exploratory factor analysis for ERP data: Insights from an inter-factor covariance decomposition. International Journal of Psychophysiology. 128; 119-136.

 Scharf F, Nestler S. (2019) A comparison of simple structure rotation criteria in temporal exploratory factor analysis for event-related potential data. Methodology. 15:43–60.

 Sussman ES. A new view on the MMN and attention debate: The role of context in processing auditory events. J Psychophysiol 2007;21:164–75. https://doi.org/10.1027/0269-8803.21.34.164.

 Wei JH, Chan TC, Luo YJ. A modified oddball paradigm ‘cross-modal delayed response’ and the research on mismatch negativity. Brain Res Bull 2002;57:221–30. https://doi.org/10.1016/S0361-9230(01)00742-0.

 Widmann A, Schröger E, Wetzel N. (2018) Emotion lies in the eye of the listener: Emotional arousal to novel sounds is reflected in the sympathetic contribution to the pupil dilation response and the P3. Biol Psychol;133:10–7.

Revisions requested by Reviewer #1:

“Similarly, please make clearer how you determined the latency of the N2b given the overlap between MMN and N2b. You explain that the MMN latency was defined as the point at which the initial negative deflection reached 60% of the peak amplitude, but it’s not clear how you determined the latency for the N2b. Which slope and which peak amplitude did you use to calculate the latency of the N2b in the absolute condition?”

We extended the explanation of the jackknife approach in the methods section (cf. ll. 300-311) and added the search windows for each ERP component both in the methods as well as the respective results sections. We used different search windows for MMN and N2b.

“Finally, please make clearer from which channels did you extract the data for your statistical comparisons. From Table 1 it looks like you picked Fz for the MMN and FCz for the N2b. Why did you choose these channels? The table also seems to show that peak difference wave amplitude for the transposed condition, for which there is potentially a clear MMN, was somewhere in the FC region (this is also confusing, see minor revisions), while the N2b in that same condition peaked somewhere in the F region. There are similar discrepancies for P3a and P3b subcomponents.” 

AND

“Table 1: Why are some of the channel labels in EP ambiguous (i.e. F instead of Fz or F1)? Why not state exactly which channels each row is referring to? Do the rows represent averaged data from multiple channels? If so which channels? Please clarify this in the caption of the table.”

We added a more elaborate explanation both to the methods section: “As there were no meaningful effects of laterality, the lateral dimension was collapsed, in order to simplify the statistical analysis. Please note that the activity values along the midline (factor frontality) represent averaged values not only including the central electrode but also the respective lateral electrodes directly adjacent to the midline electrode respectively ; e.g., the factor level frontal is the average of Fz (middle), F3 (left) and F4 (right).” (ll. 324-328) as well as a similar note in Table 1.

“General note: please make sure all your variables are italicized (namely d’s and η^2)”

We changed the formatting accordingly.

“Methods section: if it’s possible and not too much trouble, I’d appreciate a rough estimation of when the deviants actually deviated from the preceding standards (i.e. which note within the deviant stimulus was unexpected relative to the previous standard). I understand difference waves weren’t calculated this way (deviant – immediately preceding standard), but I’d like to know approximately when the average deviation occurred. You state that in the absolute condition the first note was always fixed at 400 Hz, and then you imply that the second note consistently deviated from the previous standard, but I’m not sure if I understood you correctly. This will help provide some context for the ERP latencies you’re reporting, since presumably the latencies (particularly of the early components) reflect about when the auditory and attentional systems detected a violation of the ongoing auditory regularity. If that varied a lot deviant by deviant, and if one condition varied more than the other, then the meaning of the latency differences between conditions becomes harder to interpret. At the very least I think you should provide a range of which notes could have represented deviations from the preceding standards, and perhaps address this potential issue in the discussion.”

We added the following to the discussion: “Detection of a new pattern needs at least two segments in both conditions. However, there might have been some jitter in (perceptual) change onset between conditions, resulting in a temporally more variable transposed MMN, not as well captured by a window averaging approach.” (ll.639-642). 

“Methods section: how many different tone patterns were there? Were they randomly generated or predefined?”

Patterns were randomly generated. (l. 225)

“Results section: please remind the reader of the timeframes you’re using to define each subcomponent in their subsequent “latency” section of the results”

We added the search windows for each ERP component in the respective latency results sections. MMN: l. 407f., N2b: l. 473f., P3a: l. 505f., P3b: l. 548f.

“Lines 59-60: “…facilitating the recognition of auditory objects despite large variation in spectral features”. This statement should have a reference.”

We added several references: (Bartlett & Dowling, 1980; Halpern, Bartlett, & Dowling, 1998; McDermott, Lehr, & Oxenham, 2008; Schindler, Herdener, & Bartels, 2013; Trainor, McDonald, & Alain, 2002).

“Line 69: please supply more references that speak to the “rich body of research”

We referenced several reviews that summarise research on complex sounds: Paavilainen, 2013; Peretz & Zatorre, 2005; Picton, Alain, Otten, Ritter, & Achim, 2000; Saffran & Griepentrog, 2001; Winkler, 2007).

“Line 88: “statistically robustly”. I assume you mean that there was a strong positive correlation between deviant ERP amplitude and the number of preceding standard stimuli? If that’s the case then I don’t think you need to include “statistically robustly”, as it reads as a subjective evaluation of the correlation. It’s enough to say that the amplitude increased as a function of the number of preceding stimuli.”

We dropped “robustly”.

“Lines 107-109: Please support this sentence with a reference.”

We extended this sentence and added several references: “Nonetheless, it should not be neglected that the comparison of new patterns arguably is computationally more complex when reliant on relative compared to absolute pitch information, as it is not sufficient to compare whether two pitches are the same but rather whether their relative distance is. For instance, it was found that previously heard melodies that were in the same key at exposure and test were recognized with greater accuracy than melodies that were transposed [30–33], and that the ability to process relative pitch information depends on experience [30,32,34–37].” (ll.106-114)

“Line 564: “the reported P3a amplitude decrease and latency increase…”. Fig 4a shows that P3a latencies were not different. Please clarify.”

This sentence referred to the findings by Bader et al. (2017), not to the current study. We explicitly added the reference, so there is no confusion.

“Lines 565-568: Please include references for how the P3a is associated with any of the listed stimulus features/cognitive demands.”

Although the reasoning behind these features is extensively described in the introduction section, we added some references. Now in ll. 582-583

“Line 586: “notably and significantly reduced…”. Please refrain from using subjective evaluations (notably) when statistical comparisons suffice (statistically).”

We removed the “notably” from that sentence.

Revisions requested by Reviewer #2

“A general difficulty with the approach is that there is not wide consensus in the literature as to exactly what these 4 components represent, leaving the paper with considerable speculation as to the meaning of the results.”

We tried to explicitly point out differing accounts on how to interpret modulations of said ERP components, and contrast the different inferences that could be drawn from them. While for instance, we may not be able to draw on empirically bullet-proof conceptions of these components, there is some consensus. For instance, while we may not know the exact meaning of P300 (P3b), there is considerable consensus that it reflects (i.e., covaries with) stimulus evaluation time as we described in the discussion section on P3b. 

“The literature review on MMN, P3a and relative pitch is not very complete. For example, it is stated “Since then, a rich body of research has emerged, enabling a better understanding of the processing of complex sounds”, with no references given. For example, the work of Trainor and colleagues showed MMN and P3a responses for relative pitch information in non-musicians (e.g., 2002, J Cog Neuro) and infants (e.g., He et al., 2009 Eur J Neurosci). As well, musical training may significantly modify these brain responses to relative pitch information (e.g., Fujioka et al., 2004; J Cog Neuro), as may exposure to a tonal language, so the generalizability of the results should be considered in the discussion.”

We added these references both to the introduction as well as the discussion.

“Stimuli: the stimulus patterns are very brief with the tones comprising them only 50 ms. The whole patterns are only 300 ms. Given that the temporal window of integration is 150 – 200 ms, it is not clear that these are perceived as patterns in the sense of a sequence of events. Rather they are likely processed as single units.”

We would argue that this question largely depends on the conceptualisation of what a pattern is. Our patterns are likely perceived as a consecutive stream of tone pips of different pitch (rather than as distinct notes) and in that sense probably rather as single units. Nevertheless, we know from previous studies that temporal order is well preserved in representations of such tone pip patterns (e.g. Weise, Grimm, Trujillo-Barreto & Schröger (2014) Timing matters: the processing of pitch relations, Frontiers in Neuroscience, 8:387), so the system is sensitive to the exact spectrotemporal composition of such pattern stimuli. We agree that it is an interesting question to explore whether such sound patterns are perceived as single or as concatenated units. However, in our opinion this question is beyond the scope of the current experiment. We do not think that either one of these possibilities regarding the quality of the percept is in principle incompatible with the finding that even though spectrotemporal patterns informed by relative pitch can be sufficiently represented to discriminate them well above chance from other patterns, they are processed differently compared to when absolute pitch is available.

“The jackknife approach needs more explanation. It is not clear if it was applied across participants or whether an individual latency was calculated for each participant.”

We changed the wording, to make the explanation clearer: “Specifically, the time point was estimated at which the amplitude of a particular component across leave-one-participant-out subsamples of the grand-averaged wave first reaches specific percentage values of the respective peak amplitude; slope (60%) and peak (100%).” (ll. 302-305)

“The use of when 60% and 100% of the peak of some of the components was not clear and seems arbitrary in order to make arguments that some effects of significant at 60% even though not at 100%. This needs to be justified or should be removed.”

We added the following explanation to the method section: “The 60%- relative peak estimate was included firstly, because relative latency estimates have been shown to be less noisy than peak latency estimates using the jack-knifing technique, and secondly [70], to probe whether latency effects are already present in the build-up of a given component.”

“There is a fundamental problem in separating and distinguishing the MMN and N2b components in the absolute pitch condition. Thus, without doing some kind of source localization, both the MMN and N2b results are unclear.”

We tried to address this problem in the discussion section: “However, in the context of absolute pitch there was only one prominent, quite broadly distributed frontocentral negative deflection within the typical latency range of MMN and N2b, impeding a robust estimation of MMN amplitude and latency.“ (ll. 643-646) As already described in the first section of this letter, we added a comparative figure to the supplements. This includes both the data from the current study as well as the data from Bader et al. (2017) to contrast the ERPs from active and passive listening. We argue that this figure shows that the large negative deflection is only present within the active listening task, and that there is a consistent time course for the initial negative deflection compatible with MMN elicitation in both active and passive listening and both absolute and relative pitch conditions respectively.

“MMN was actually found to be a bit larger in absolute condition compared to the relative condition, which is actually not in agreement with Bader et al. However, this is inconclusive as, with the current analysis methods, it was not really possible to separate MMN and N2b. The authors argue that because of this MMN is not really different across conditions, but I think this remains unknown.”

Given the more central contributions in the absolute compared to the transposed MMN, we argued that the likeliest explanation would be a spatial and temporal overlap by N2b in the absolute condition. Naturally, we cannot preclude that the significant difference in the MMN window reflects a true MMN amplitude difference. However, as already pointed out, the strong overlap with N2b did not allow a robust MMN amplitude (& latency) estimation. As it is well known that N2b overlaps with MMN in active paradigms, (cf. introduction and discussion for references) we would still argue this is the far likelier explanation. Especially given that Bader et al. (2017) did not find a significant MMN amplitude difference in passive listening. Of course, absence of evidence is not evidence of absence. Therefore, we extended this discussion part to more explicitly explain the interpretations of a potential true amplitude difference (ll. 629ff.).

“With respect to N2b it was found to be later in the relative than absolute pitch condition. However, this is not completely convincing as the MMN and N2b could not be completely separated. This delayed N2b for relative pitch is interpreted as increased discrimination demands, but it seems that other interpretations might be possible.”

We tried to discuss several different interpretations (ll. 699ff.): “Taken together, it seems that N2b latency not merely indexes the speed but also the complexity, and perhaps accuracy, of the processing of deviating and, or target stimulus characteristics. In these terms, the increase in latency in the relative compared with the absolute pitch context, might well reflect generally increased computational complexity when a pattern is defined by relative rather than absolute pitch. As Ritter et al. (1992) pointed out that the interpretation of N2b largely relies “on the experimental circumstances and the strategies used by the subjects”, one could argue even further: N2b latency differences as a function of pitch might not just reflect differences in mere processing time between absolute and relative patterns, but might actually indicate wholly different kind of processes operating at the stage of stimulus comparison or categorisation respectively [97,100].”

“The amplitude of the N2b was “descriptively” smaller in amplitude for the relative pitch case, but not statistically different. A somewhat convoluted argument is made that N2b might reflect both “representational strength” and “attention” and that the lack of significant amplitude difference might reflect some combination of reduced “representational strength” and increases “attention in the relative pitch case. This seems overly speculative.”

We did not mean that N2b amplitude is necessarily modulated by both. As findings on N2b amplitude modulations are generally rather inconsistent, we aimed to stress that there are several different explanations that could account for the reported findings. Some of them could relate to differences in the processing of absolute and relative pitch at the level of N2b, but such a descriptive difference could just as well be accounted by the differential overlap with P3a – as the N2b peaked later in the relative pitch context, it more temporally overlapped with P3a (which did not show a significant latency effect by pitch context).

“No amplitude or latency differences were found for P3a for relative and absolute conditions. This is consistent with the idea that top-down processes (attention?) compensate for any increased difficult of the relative pitch condition. But the authors go on to state the absolute and relatively pitch information might just be processed “differently”. I couldn’t follow this argument.”

We meant to say that in active listening we did not find evidence that processing of true pattern changes is affected by absence of absolute pitch information at the level of P3a. However, we would advise caution to infer from this finding in active listening, that the same holds true for passive listening. Simply because top-down processes could compensate in active listening, the same might not be true in passive listening. Such an inference is, in our opinion, beyond the scope of the current study. During the comparison of the current data with Bader et al. (2017) we noticed that when taking the data from both studies together, this point is stressed even further. We added this to the discussion in the following manner: “Actually, in the direct comparison of passive and active listening task (supporting information S3), it appears that P3a elicited by true pattern changes might be separable into two distinct subcomponents [95–99] – namely an early and a late P3a. When visually comparing both studies a potential late P3a was elicited of similar latency and amplitude in both studies irrespective of pitch information. However, a potential early P3a was elicited only in response to absolute true pattern changes in passive listening. Thus, this early P3a activity could be a candidate to further elucidate differential pattern processing as a function of pitch in future studies.” Thus, in passive listening absence of absolute pitch information might not have caused an increase in P3a latency per se, but rather early P3a seems to only be elicited in response to absolute and not relative true pattern changes. Such an early P3a was not observed in active listening regardless of type of pitch information. This points to a striking divergence between passive and active processing of absolute and relative pitch.

“From the results, the N2b and P3b were both delayed in the relative compared to absolute condition. The authors interpret the N2b as representing “target selection” and the P3b as representing change detection, or working memory or response related processing. Therefore, they conclude that target selection is delayed from relative pitch, but that change detection, working memory and response-related processing are not of themselves delayed. This argument of course depends on accurate assessment of N2b latency.”

To clarify, we would argue that P3b reflects latency stimulus evaluation time in the sense that it covaries with it, but does not represent change detection, working memory processes itself – which is in line with literature (e.g. Dien, 2014; Verleger, 2020). This means, whatever time penalty relative compared to absolute pitch introduces, likely happened at previous processing stages. We did not find evidence that MMN or (late) P3a were delayed. We revised the discussion section on P3b as follows to clarify: “Within the traditional framework [51,52] the lack of evidence for modulation of P3b amplitude by pitch information implies that during the revision, or updating of the current mental model in response to the detection of a true pattern change, attentional resources are equally available [22,38]. A more recent account suggests that P3b rather reflects reactivation of stimulus-response links, and that P3b amplitude increases as a function of response infrequency [108]. The absence of P3b amplitude modulation in the current study would thus be explained by the fact that true pattern changes occurred equiprobably in both absolute and relative pitch context, resulting in comparable response frequencies. Within both contexts, the increase in P3b latency would signify increased stimulus evaluation time [22,48,109]. Indeed, the observed delay was already present at the N2b level, and did not notably increase at the level of P3b. That P3b peaked after the correct behavioural response was made [24], implies that it reflects post-identification stimulus categorisation, response selection and execution processes [24,48,49,55,56,108] which are not vulnerable to the absence of pitch information.” (ll. 752-765). We agree that this inference rests on the reported N2b latency shift.

“I find it curious that in the conclusions, no mention is made of MMN or P3a. There is no discussion of how these two component relate to N2b and P3b in a stages of processing account. And why, for example, N2b can be delay while the following P3a is not delayed.”

We added a sentence on MMN and P3a in the conclusion section. In terms of processing stages, we connected MMN to sensory change detection, N2b to target selection, P3a to working memory, P3b to (post-) response related-processes. We are not aware of literature providing a robust picture of how these components are linked and interact. We therefore decided against engaging in a speculative attempt to explain this.

Yours sincerely,

Nina Coy (on behalf of all co-authors)

---

## [Decision Letter · Decision Letter 1]

19 Jan 2021

PONE-D-20-25390R1

Change detection of auditory tonal patterns defined by absolute versus relative pitch information. A combined behavioural and EEG study.

PLOS ONE

Dear Dr. Coy,

Thank you for submitting your manuscript to PLOS ONE. After careful consideration, we feel that it has merit but does not fully meet PLOS ONE’s publication criteria as it currently stands. Therefore, we invite you to submit a revised version of the manuscript that addresses the points raised during the review process.

We look forward to receiving your revised manuscript.

Kind regards,

Qian-Jie Fu, Ph.D.

Academic Editor

PLOS ONE

Reviewers' comments:

Reviewer's Responses to Questions

**Comments to the Author**

1. If the authors have adequately addressed your comments raised in a previous round of review and you feel that this manuscript is now acceptable for publication, you may indicate that here to bypass the “Comments to the Author” section, enter your conflict of interest statement in the “Confidential to Editor” section, and submit your "Accept" recommendation.

Reviewer #1: (No Response)

2. Is the manuscript technically sound, and do the data support the conclusions?

Reviewer #1: Yes

3. Has the statistical analysis been performed appropriately and rigorously? 

Reviewer #1: Yes

4. Have the authors made all data underlying the findings in their manuscript fully available?

Reviewer #1: Yes

5. Is the manuscript presented in an intelligible fashion and written in standard English?

Reviewer #1: Yes

6. Review Comments to the Author

Reviewer #1: The authors have thoroughly addressed all my comments. I believe the concern about separation of MMN and N2b is most adequately addressed in Fig 3 of the response to reviewers, which directly compares the ERPs in active and passive paradigms. This figure, and some of the preceding discussion, should be included in the paper itself as further evidence of the temporal overlap between MMN and N2b components in the absolute condition. I would also recommend making clearer that the ERPs shown in column 3 are difference waves, and which waveforms are from the active and which are from the passive condition, in both the legend and the caption. I eventually figured it out but it took me a while. Otherwise I think the paper is much better, much clearer, and is ready for publication.

7. PLOS authors have the option to publish the peer review history of their article (what does this mean?). If published, this will include your full peer review and any attached files.

Reviewer #1: No

---

## [Author Response · Author response to Decision Letter 1]

29 Jan 2021

Dear Reviewers,

We are pleased to resubmit for publication the revised version of the manuscript entitled “Change detection of auditory tonal patterns defined by absolute versus relative pitch information. A combined behavioural and EEG study.” by Nina Coy, Maria Bader, Erich Schröger, and Sabine Grimm.

We appreciate the positive response to our first revision. We have amended the requested changes from Reviewer #1 as follows.

Reviewer #1: “The authors have thoroughly addressed all my comments. I believe the concern about separation of MMN and N2b is most adequately addressed in Fig 3 of the response to reviewers, which directly compares the ERPs in active and passive paradigms. This figure, and some of the preceding discussion, should be included in the paper itself as further evidence of the temporal overlap between MMN and N2b components in the absolute condition. I would also recommend making clearer that the ERPs shown in column 3 are difference waves, and which waveforms are from the active and which are from the passive condition, in both the legend and the caption. I eventually figured it out but it took me a while. Otherwise I think the paper is much better, much clearer, and is ready for publication.“

We have now included the, previously supplementary, figure comparing the ERP data between active and passive listening to the main manuscript body (now Fig 5) and extended its discussion. We changed the legend and added some more explicit information to improve its readability, especially concerning the differentiation between stimulus level and difference level ERPs.

Yours sincerely,

Nina Coy (on behalf of all co-authors)

---

## [Editor Report · Decision Letter 2]

9 Feb 2021

Change detection of auditory tonal patterns defined by absolute versus relative pitch information. A combined behavioural and EEG study.

PONE-D-20-25390R2

Dear Dr. Coy,

We’re pleased to inform you that your manuscript has been judged scientifically suitable for publication and will be formally accepted for publication once it meets all outstanding technical requirements.

Kind regards,

Qian-Jie Fu, Ph.D.

Academic Editor

PLOS ONE
---

## [Editor Report · Acceptance letter]

17 Feb 2021

PONE-D-20-25390R2 

Change detection of auditory tonal patterns defined by absolute versus relative pitch information. A combined behavioural and EEG study. 

Dear Dr. Coy:

I'm pleased to inform you that your manuscript has been deemed suitable for publication in PLOS ONE. Congratulations! Your manuscript is now with our production department. 

Kind regards, 

on behalf of

Dr. Qian-Jie Fu 

Academic Editor

PLOS ONE